# Global Reward Maximization with Partial Feedback via Differentiallly Private Distributed Linear Bandits

## Abstract

In this paper, we study the problem of global reward maximization with only partial distributed feedback. This problem is motivated by several real-world applications (e.g., cellular network configuration, dynamic pricing, and policy selection) where an action was taken by a central entity that influences a large population that contributes to the global reward. However, collecting such reward feedback from the entire population not only incurs a prohibitively high cost, but often leads to privacy concerns. To tackle this, we formulate it as a differentially private distributed linear bandits (DP-DLB), where only a subset of users from the population are selected (called clients) to participate in the learning process and the central server learns the global model from such partial feedback by iteratively aggregating these clients' local feedback in a differentially private fashion. We then propose a unified algorithmic learning framework, called differentially private distributed phased elimination (DP-DPE), which enables us to naturally integrate popular differential privacy (DP) models (including central DP, local DP, and shuffle DP) into the learning process. Furthermore, we analyze the performance of the DP-DPE algorithm and show that DP-DPE achieves both sublinear regret and sublinear communication cost. Interestingly, we highlight that DP-DPE allows us to achieve privacy protection "for free" as the additional cost due to privacy can be a lower-order additive term. Finally, we conduct simulations to corroborate our theoretical results and demonstrate the effectiveness of DP-DPE in terms of regret, communication cost, and privacy guarantees.

## 1 Introduction

The bandit learning models have been widely adopted for many sequential decision-making problems, such as clinical trials, recommender systems, and configuration selection. Each action (called arm), if selected in a round, generates a (noisy) reward. By observing such reward feedback, the learning agent gradually learns the unknown parameters of the model (e.g., mean rewards) and decides the action in the next round. The objective of the learning agent is to maximize the cumulative reward over a finite time horizon, balancing the tradeoff between *exploitation* and *exploration*. While the stochastic multi-armed bandits (MAB) model has proven to be useful for these applications (Lai & Robbins, 1985), one key limitation is that actions are assumed to be independent, which, however, is usually not the case in practice. Therefore, the linear bandit model that captures the correlation among actions has been extensively studied (Lattimore & Szepesvári, 2020; Abbasi-Yadkori et al., 2011; Li et al., 2010).

In this paper, we introduce a new linear bandit setting where the reward of an action could be from a large population. Take the cellular network configuration as an example (see Fig. 1). The configuration (antenna tilt, maximum output power, inactivity timer, etc.) of a base station (BS), denoted by $x \in \mathbb{R}^d$, influences all the users under the coverage of this BS (Mahimkar et al., 2021). Once a certain configuration is set, the BS receives a reward in terms of the network-level performance, which accounts for the performance of all users within the coverage (e.g., average user throughput). Specifically, let the mean global reward of configuration $x$ be $f(x) = \langle \theta^*, x \rangle$, where $\theta^* \in \mathbb{R}^d$ represents the unknown global parameter. While a certain configuration may work best for a specific user, only one configuration can be applied at the BS at a time,

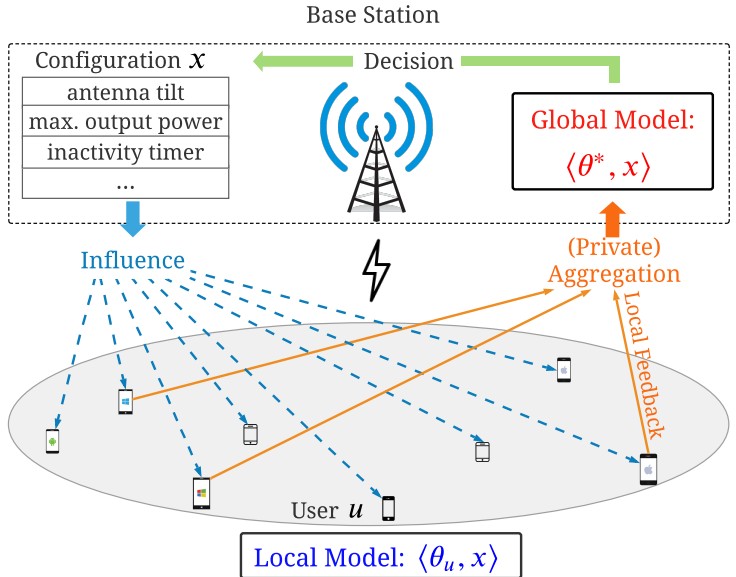

Figure 1: Cellular network configuration: a motivating application of global reward maximization with partial feedback in a distributed linear bandit setting.

which, however, simultaneously influences all the users under the coverage. Therefore, the goal here is to find the best configuration that maximizes the global reward (i.e., the network-level performance).

At first glance, it seems that one can address the above problem by applying existing linear bandit algorithms (e.g., LinUCB (Li et al., 2010)) to learn the global parameter $\theta^*$. However, this would require collecting reward feedback from the entire population, which could incur a prohibitively high cost or could even be impossible to implement in practice when the population is large. To learn the global parameter, one natural way is to sample a subset of users from the population and aggregate these distributed partial feedback. This leads to a new problem we consider in this paper: *global reward maximization with partial feedback in a distributed linear bandit setting.* As in many distributed supervised learning problems (Bassily et al., 2019; Geyer et al., 2017; Girgis et al., 2021), privacy protection is also of significant importance in our setting as clients' local feedback may contain their sensitive information. In summary, we are interested in the following fundamental question:

*How to privately achieve global reward maximization with only partial distributed feedback?*

To tackle this, we introduce a *differentially private distributed linear bandit (DP-DLB)* formulation. In DP-DLB, there is a global linear bandit model $f(x) = \langle \theta^*, x \rangle$ with an unknown parameter $\theta^* \in \mathbb{R}^d$ at the central server (e.g., the BS); each user $u$ of the population has a local linear bandit model $f_u(x) = \langle \theta_u, x \rangle$, which represents the mean local reward for user $u$. Here, we assume that each user $u$ is associated with a different local parameter $\theta_u \in \mathbb{R}^d$. This is motivated by the fact that the mean local reward (e.g., the expected throughput of a user under a certain network configuration) varies across the users. In addition, each $\theta_u$ is an unknown local parameter and is assumed to be a local realization of a random vector with the mean being the global model parameter $\theta^*$. The server makes decisions based on the estimated global model, which can be learned through sampling a subset of users (referred to as clients) and iteratively aggregating these distributed partial feedback. While sampling more clients could improve the learning accuracy and thus lead to a better performance, it also incurs a higher communication cost. Therefore, it is important to address this tradeoff in the design of communication protocols. Furthermore, to protect users' privacy, we resort to *differential privacy (DP)* to guarantee that clients' sensitive information will not be inferred by an adversary. Therefore, the ultimate goal in DP-DLB is to maximize the cumulative global reward (or equivalently minimize the regret due to not choosing the optimal action in hindsight) while minimizing the

Table 1: Summary of our main results

| Algorithm[1] | Regret[2] | Communication cost[3] |
|---|---|---|
| DPE | $O\left(T^{1-\alpha/2}\sqrt{d\log(kT)}\right)$ | $O(dT^{\alpha})$ |
| CDP-DPE | $O\left(T^{1-\alpha/2}\sqrt{d\log(kT)} + T^{1-\alpha}d^{1+\alpha}\sqrt{\ln(1/\delta)}\sqrt{\log(kT)}/\epsilon\right)$ | $O(dT^{\alpha})$ |
| LDP-DPE | $O\left(T^{1-\alpha/2}\sqrt{d\log(kT)} + T^{1-\alpha/2}d^{3/2}\sqrt{\ln(1/\delta)}\sqrt{\log(kT)}/\epsilon\right)$ | $O(dT^{\alpha})$ |
| SDP-DPE | $O\left(T^{1-\alpha/2}\sqrt{d\log(kT)} + T^{1-\alpha}d^{1+\alpha}\ln(d/\delta)\sqrt{\log(kT)}/\epsilon\right)$ | $O(dT^{3\alpha/2})$ (bits) |

[1]DPE is the non-private DP-DPE algorithm; CDP-DPE, LDP-DPE, and SDP-DPE represent the DP-DPE algorithm in the central, local, and shuffle models, respectively, which guarantee $(\epsilon, \delta)$-DP, $(\epsilon, \delta)$-LDP, and $(\epsilon, \delta)$-SDP, respectively.

[2]In the regret upper bounds, $T$ is the length of the time horizon, $k$ is the number of actions, $d$ is the dimension of the action space, and $\alpha$ is a design parameter that can be used to tune the tradeoff between the regret and the communication cost.

[3]While the communication cost of CDP-DPE and LDP-DPE is measured in the number of real numbers transmitted between the clients and the server, SDP-DPE directly uses bits for reporting feedback. A detailed discussion is provided in Section 4.

communication cost and providing privacy guarantees for the participating clients. Our main contributions are summarized as follows.

- To the best of our knowledge, this is the first work that considers global reward maximization with partial feedback in the distributed linear bandit setting. In addition to the traditional tradeoff between exploitation and exploration, learning with such partial distributed feedback introduces two practical challenges: communication efficiency and privacy concerns, which add an extra layer of difficulty in the design of learning algorithms.

- To address these challenges, we introduce a novel DP-DLB formulation and develop a carefully-crafted algorithmic learning framework, called differential private distributed phased elimination (DP-DPE), which allows the server and the clients to work in concert and also naturally integrates several state-of-the-art DP trust models (including central model, local model, and shuffle model). This unified framework enables us to systemically study the key regret-privacy-communication tradeoff.

- Then, we establish the regret-privacy-communication tradeoff of DP-DPE in various settings including the non-private case as well as the central, local, and shuffle DP models. Our main results are as summarized in Table 1. These results reveal that DP-DPE enables us to achieve privacy "for-free" in the central and shuffle models, in the sense that the additional regret due to privacy protection is only a lower-order additive term. Moreover, to the best of our knowledge, this is the first work that considers the shuffle model in linear bandits to attain a better regret-privacy tradeoff, i.e., a similar privacy protection as the strong local model while achieving the same regret as the central model. We further perform simulations on synthetic data to corroborate our theoretical results.

- Finally, we observe an interesting connection between our introduced DP-DLB formulation and the differentially private stochastic convex optimization (DP-SCO) problem in terms of achieving privacy "for-free". This bridge between our online bandit learning and the standard supervised learning might be of independent interest.

The rest of the paper is organized as follows. We first present the system model and problem formulation in Section 2. Then, we describe the algorithmic learning framework, DP-DPE, in Section 3, followed by its instanstiations with different DP models in Section 4. In Section 5, we provide the performance analysis for different DP-DPE instantiations, followed by discussions on achieving privacy "for free" as well as on the connection between our introduced DP-DLB formulation and the DP-SCO problem in Section 6. Our numerical results are presented in Section 7. Finally, we discuss the related work in Section 8 and make concluding remarks in Section 9.

## 2 System Model and Problem Formulation

We begin with some basic notations: $[N] \triangleq \{1, \ldots, N\}$ for any positive integer $N$; $|S|$ denotes the cardinality of set $S$; $\|x\|_2$ denotes the $\ell_2$-norm of vector $x$; the inner product is denoted by $\langle \cdot, \cdot \rangle$. For a positive definite matrix $A \in \mathbb{R}^{d \times d}$, the weighted $\ell_2$-norm of vector $x \in \mathbb{R}^d$ is defined as $\|x\|_A \triangleq \sqrt{x^\top A x}$. For any sequence $\{a_t\}_{t=1}^\infty$, we use $a_{i:j}$ to denote the subsequence $a_i, \ldots, a_j$. We use $\mathcal{A} \circ \mathcal{B}$ to represent the composition of algorithms (or functions) $\mathcal{A}$ and $\mathcal{B}$.

### 2.1 Global Reward Maximization with Partial Feedback

We consider the global reward maximization problem over a large population, which is a sequential decision making problem. In each round $t$, the learning agent (e.g., the BS or the policy maker) selects an action $x_t$ from a finite decision set $\mathcal{D} \subseteq \{x \in \mathbb{R}^d : \|x\|_2^2 \leq 1\}$ with $|\mathcal{D}| = k$. This action leads to a global reward with mean $\langle \theta^*, x_t \rangle$, where $\theta^* \in \mathbb{R}^d$ with $\|\theta^*\|_2 \leq 1$ is unknown to the agent. This global reward captures the overall effectiveness of action $x_t$ over the entire population $\mathcal{U}$. The local reward of action $x_t$ at user $u$ has a mean $\langle \theta_u, x_t \rangle$ with $\theta_u \in \mathbb{R}^d$, where $\theta_u$ is the local parameter, which is assumed to be a realization of a random vector with mean $\theta^*$ and is also unknown. Let $x^* \triangleq \arg\max_{x \in \mathcal{D}} \langle \theta^*, x \rangle$ be the unique global optimal action. Then, the objective of the agent is to maximize the cumulative global reward, or equivalently, to minimize the regret defined as follows:

$$R(T) \triangleq T \langle \theta^*, x^* \rangle - \sum_{t=1}^T \langle \theta^*, x_t \rangle. \tag{1}$$

At first glance, standard linear bandit algorithms (e.g., LinUCB in Li et al. (2010)) can be applied to addressing the above problem. However, the exact reward here is a global quantity, which is the average over the entire population. The learning agent may not be able to observe this exact reward, since collecting such global information from the entire population incurs a prohibitively high cost and could often be impossible to implement in practice.

### 2.2 Differentially Private Distributed Linear Bandits (DP-DLB)

To address the above problem, we consider a *differentially private distributed linear bandit (DP-DLB)* formulation, where there are two important entities: a central server (which wants to learn the global model) and participating clients (i.e., a subset of users from the population who are willing to share their feedback). In the following, we discuss important aspects of the DP-DLB formulation.

**Server.** The server is involved in learning the global linear bandit model, i.e., unknown parameter $\theta^*$. In each round $t$, it selects an action $x_t$ with the objective of maximizing the cumulative global reward $\sum_{t=1}^T \langle \theta^*, x_t \rangle$. Without direct observation of the exact reward of action $x_t$, the server only collects partial feedback from a subset of users sampled from the population, called *clients*, and then aggregates these partial distributed feedback to learn the global parameter $\theta^*$. Based on the learned model, the server chooses an action in the next round.

**Clients.** We assume that each participating client is randomly sampled from the population and is independent from each other and also from other randomness. Specifically, we assume that the local parameter $\theta_u$ at client $u$ satisfies $\theta_u = \theta^* + \xi_u$, where $\xi_u \in \mathbb{R}^d$ is a zero-mean $\sigma$-sub-Gaussian random vector[4] and is independently and identically distributed (*i.i.d.*) across all clients. Let $U_t$ be the set of participating clients in round $t$. After action $x_t$ is chosen by the server in round $t$, each client $u \in U_t$ observes a noisy local reward: $y_{u,t} = \langle \theta_u, x_t \rangle + \eta_{u,t}$, where $\eta_{u,t}$ is a conditionally 1-sub-Gaussian[5] noise and *i.i.d.* across the clients and also over time. We also assume that the local rewards are bounded, i.e., $\|y\|_2 \leq B$, for all $u \in \mathcal{U}$ and $t \in [T]$.

---

[4]A random vector $\xi \in \mathbb{R}^d$ is said to be $\sigma$-sub-Gaussian if $\mathbb{E}[\xi] = 0$ and $v^\top \xi$ is $\sigma$-sub-Gaussian for any unit vector $v \in \mathbb{R}^d$ and $\|v\|_2 = 1$ (Bühlmann & Van De Geer, 2011).

[5]Consider noise sequence $\{\eta_t\}_{t=1}^\infty$. As in the general linear bandit model (Lattimore & Szepesvári, 2020), $\eta_t$ is assumed to be conditionally 1-sub-Gaussian, meaning $\mathbb{E}[e^{\lambda \eta_t} | x_{1:t}, \eta_{1:t}] \leq \exp(\lambda^2/2)$ for all $\lambda \in \mathbb{R}$.

**Communication.** The communication happens when the clients report their feedback to the server. At the beginning of each communication step, each participating client reports feedback to the server based on the local reward observations during a certain number of rounds. In particular, the time duration between reporting feedback is called a phase. By aggregating such feedback from the clients, the server estimates the global parameter $\theta^*$ and adjusts its decisions in the following rounds accordingly. Suppose that the participating clients do not quit during a phase. By slightly abusing the notation, we also use $U_l$ to denote the set of participating clients in the $l$-th phase.

Note that communication cost is a critical factor in the DP-DLB framework. We use the same definition of the communication cost as in Wang et al. (2019): the total number of real numbers (or bits) communicated between the server and the participating clients. Let $L$ be the total number of phases in $T$ rounds, and let $N_l$ be the number of real numbers (or bits) communicated in the $l$-th phase. Then, the total communication cost, denoted by $C(T)$, can be calculated as follows:

$$C(T) \triangleq \sum_{l=1}^{L} |U_l| N_l. \tag{2}$$

**Data privacy.** In practice, even if users are willing to share their feedback, they typically require privacy protection as a premise. This requires our learning algorithm to guarantee that clients' sensitive information will not be revealed during the learning process. To this end, we will resort to *differential privacy (DP)* (Dwork et al., 2006) to formally address the privacy concerns in the learning process. More importantly, instead of only considering the standard central model where the central server is responsible for protecting the privacy, we would also like to incorporate other popular DP models in a unified way, including the stronger local model (where each client directly protects her data) (Kasiviswanathan et al., 2011) and the recently proposed shuffle model (where a trusted shuffler in-between clients and server is adopted to amplify privacy) (Cheu et al., 2019). We provide the formal definitions of different DP models in Section 4.

## 3 Algorithm Design

In this section, we first present the key challenges within the introduced DP-DLB formulation and then describe the developed DP-DPE framework to resolve these challenges.

### 3.1 Key Challenges

To solve the problem of global reward maximization with partial feedback using the DP-DLB formulation, we face the following new challenges.

As in the standard stochastic bandit problem, there is uncertainty due to noisy rewards of each chosen action, which is called the *action-related uncertainty*. In addition to this, we face another type of uncertainty related to the sampled clients in DP-DLB, which is called the *client-related uncertainty*. The client-related uncertainty lies in estimating the global model at the server based on randomly sampled clients with *biased* local models. Note that the global model may not be accurately estimated even if exact rewards of the sampled clients are known when the number of sampled clients is insufficient. Therefore, the first challenge lies in *simultaneously addressing both types of uncertainty in a sample-efficient way* (Challenge ⓐ).

To handle the newly introduced client-related uncertainty, we must sample a sufficiently large number of clients so that the global parameter can be accurately estimated using the partial distributed feedback. However, too many clients result in a large communication cost (as defined in Eq. (2)). Therefore, the second challenge is to *decide the number of sampled clients for addressing the tradeoff between the regret (due to the client-related uncertainty) and the communication cost* (Challenge ⓑ).

Finally, to ensure privacy guarantees for the participating clients, one needs to add additional perturbations (or noises) to the local feedback. *Such randomness introduces another type of uncertainty to the learning process* (Challenge ⓒ), and *it is unclear how to integrate different trust models and DP mechanisms into a unified algorithmic learning framework* (Challenge ⓓ). These add an extra layer of difficulty in the design of learning algorithms.

**Main ideas.** To address the aforementioned challenges ⓐ-ⓓ, we present our main ideas as follows. Considering the communication scheme, we are interested in a phased elimination algorithm that gradually eliminates suboptimal actions by periodically aggregating and analyzing the local feedback from the sampled clients in a privacy-preserving manner. To address the multiple types of uncertainty when estimating the global reward (ⓐⓒ), we carefully construct a confidence width to incorporate all three types of uncertainty. To achieve a sublinear regret while minimizing communication cost (ⓑ), we increase both the phase length and the number of participating clients exponentially. To ensure privacy guarantees (ⓓ), we introduce a notion, called PRIVATIZER, which can be easily tailored under different DP models. Note that the PRIVATIZER is a process consisting of tasks to be collaboratively completed by the clients, the server, and/or even a trusted third party. To keep this notion general, we use $\mathcal{P} = (\mathcal{R}, \mathcal{S}, \mathcal{A})$ to denote a PRIVATIZER, where $\mathcal{R}$ is the procedure at each client (usually a local randomizer), $\mathcal{S}$ is a trusted third party that helps privatize data (e.g., a shuffler that permutes received messages), and $\mathcal{A}$ is an analyzer operated by the central server. In the following, we will show how to integrate all these ideas into a unified framework.

## 3.2 Differentially Private Distributed Phased Elimination (DP-DPE)

With the main ideas presented above, we propose a unified algorithmic learning framework, called *differentially private distributed phased elimination (DP-DPE)*, which is built on the (non-private) phased elimination algorithm for traditional stochastic linear bandits (Lattimore & Szepesvári, 2020; Lattimore et al., 2020).

We present DP-DPE in Algorithm 1, which runs in phases and operates with the coordination of the central server and the participating clients in a synchronized manner. At a high level, each phase consists of the following three steps:

1) **Action selection (Lines 4-6):** computing a near-$G$-optimal design (i.e., a distribution) over a set of possibly optimal actions and playing these actions;

2) **Clients sampling and private feedback aggregation (Lines 7-16):** sampling participating clients and aggregating their local feedback in a privacy-preserving fashion;

3) **Parameter estimation and action elimination (Lines 17-19):** using (privately) aggregated data to estimate $\theta^*$ and eliminating actions that are likely to be suboptimal.

In the following, we describe the detailed operations of DP-DPE. We begin by giving some additional notations. Consider the $l$-th phase. Let $t_l$ and $T_l$ be the index of the starting round and the length of the $l$-th phase, respectively. Then, let $\mathcal{T}_l \triangleq \{t \in [T] : t_l \leq t < t_l + T_l\}$ be the round indices in the $l$-th phase, let $\mathcal{T}_l(x) \triangleq \{t \in \mathcal{T}_l : x_t = x\}$ be the time indices in the $l$-th phase when action $x$ is selected, and let $\mathcal{D}_l \subseteq \mathcal{D}$ be the set of active actions in the $l$-th phase.

**Action selection (Lines 4-6):** In the $l$-th phase, the action set $\mathcal{D}_l$ consists of active actions that are possibly optimal. We compute a distribution $\pi_l(\cdot)$ over $\mathcal{D}_l$ and choose actions according to $\pi_l(\cdot)$. We briefly explain the intuition below. Let $V(\pi) \triangleq \sum_{x \in \mathcal{D}} \pi(x) x x^\top$ and $g(\pi) \triangleq \max_{x \in \mathcal{D}} \|x\|^2_{V(\pi)^{-1}}$. According to the analysis in Lattimore & Szepesvári (2020, Chapter 21), if action $x \in \mathcal{D}$ is played $\lceil h\pi(x) \rceil$ times (where $h$ can be an arbitrary positive constant), the estimation error associated with the action-related uncertainty for action $x$ is at most $\sqrt{2g(\pi) \log(1/\beta)/h}$ with probability $1 - \beta$ for any $\beta \in (0, 1)$. That is, for a fixed number of rounds, a distribution $\pi(\cdot)$ with a smaller value of $g(\pi)$ helps achieve a better estimation. Note that minimizing $g(\cdot)$ is a well-known *G-optimal design* problem (Pukelsheim, 2006). According to the Kiefer-Wolfowitz Theorem (Kiefer & Wolfowitz, 1960), one can find a distribution $\pi^*$ minimizing $g(\cdot)$ with $g(\pi^*) = d$, and the support set[6] of $\pi^*$, denoted by supp($\pi^*$), has a size no greater than $d(d+1)/2$. In our problem, however, it suffices to solve it near-optimally, i.e., finding a distribution $\pi_l$ such that $g(\pi_l) \leq 2d$ with $|\text{supp}(\pi_l)| \leq 4d \log \log d + 16$ (Line 4), which follows from Lattimore et al. (2020, Proposition 3.7). The near-$G$-optimal design reduces complexity to $O(kd^2)$ while keeping the same order of regret.

---

[6] The support set of a distribution $\pi$ over set $\mathcal{D}$, denoted by $\text{supp}_{\mathcal{D}}(\pi)$, is the subset of elements with a nonzero $\pi(\cdot)$, i.e., $\text{supp}_{\mathcal{D}}(\pi) \triangleq \{x \in \mathcal{D} : \pi(x) \neq 0\}$. We drop the subscript $\mathcal{D}$ in $\text{supp}_{\mathcal{D}}(\pi)$ and use $\text{supp}(\pi)$ for notational simplicity.

---

**Algorithm 1** Differentially Private Distributed Phased Elimination (DP-DPE)

---

1: **Input:** $\mathcal{D} \subseteq \mathbb{R}^d$, $\alpha \in (0,1)$, and confidence level $\beta \in (0,1)$
2: **Initialization:** $l = 1$, $t_1 = 1$, $\mathcal{D}_1 = \mathcal{D}$, and $h_1 = 4d \log \log d + 16$
3: **while** $t_l \leq T$ **do**
4:   Find a distribution $\pi_l(\cdot)$ over $\mathcal{D}_l$ such that $g(\pi_l) \triangleq \max_{x \in \mathcal{D}_l} \|x\|^2_{V(\pi_l)^{-1}} \leq 2d$
    and $|\text{supp}(\pi_l)| \leq 4d \log \log d + 16$, where $V(\pi_l) \triangleq \sum_{x \in \mathcal{D}_l} \pi_l(x) x x^\top$
5:   Let $T_l(x) = \lceil h_l \pi_l(x) \rceil$ for each $x$ in $\text{supp}(\pi_l)$ and $T_l = \sum_{x \in \text{supp}(\pi_l)} T_l(x)$
6:   Play each action $x \in \text{supp}(\pi_l)$ exactly $T_l(x)$ times if not reaching $T$
7:   Randomly select $\lceil 2^{\alpha l} \rceil$ participating clients $U_l$
    # Operations at each client
8:   **for** each client $u \in U_l$ **do**
9:     **for** each action $x \in \text{supp}(\pi_l)$ **do**
10:       Compute its average local observations $y_l^u(x)$ in $T_l(x)$ rounds, i.e.,
      $y_l^u(x) = \frac{1}{T_l(x)} \sum_{t \in \mathcal{T}_l(x)} (\langle \theta_u, x \rangle + \eta_{u,t})$
11:     **end for**
12:     Let $\vec{y}_l^u = (y_l^u(x))_{x \in \text{supp}(\pi_l)} \in \mathbb{R}^{|\text{supp}(\pi_l)|}$
      # Apply the PRIVATIZER $\mathcal{P} = (\mathcal{R}, \mathcal{S}, \mathcal{A})$
      # The local randomizer $\mathcal{R}$ at each client:
13:     Run the local randomizer $\mathcal{R}$ and send the output $\mathcal{R}(\vec{y}_l^u)$ to the shuffler $\mathcal{S}$
14:   **end for**
    # The shuffler $\mathcal{S}$ at a trusted third party:
15:   Run the shuffler $\mathcal{S}$ to uniformly permute messages and send the output $\mathcal{S}(\{\mathcal{R}(\vec{y}_l^u)\}_{u \in U_l})$ to the analyzer $\mathcal{A}$
    # The analyzer $\mathcal{A}$ at the server:
16:   Generate the privately aggregated statistics: $\tilde{y}_l = \mathcal{A}(\mathcal{S}(\{\mathcal{R}(\vec{y}_l^u)\}_{u \in U_l}))$
17:   Compute the following quantities:

$$\begin{cases} V_l = \sum_{x \in \text{supp}(\pi_l)} T_l(x) x x^\top \\ G_l = \sum_{x \in \text{supp}(\pi_l)} T_l(x) x \tilde{y}_l(x) \\ \tilde{\theta}_l = V_l^{-1} G_l \end{cases}$$

18:   Find low-rewarding actions based on confidence width $W_l$:

$$E_l = \left\{ x \in \mathcal{D}_l : \max_{b \in \mathcal{D}_l} \langle \tilde{\theta}_l, b - x \rangle > 2W_l \right\}$$

19:   Update $\mathcal{D}_{l+1} = \mathcal{D}_l \backslash E_l$; $h_{l+1} = 2h_l$; $t_{l+1} = t_l + T_l$; $l = l + 1$
20: **end while**

---

**Clients sampling and private feedback aggregation (Lines 7-16):** The central server randomly samples a subset $U_l$ of $\lceil 2^{\alpha l} \rceil$ users (called clients) from $\mathcal{U}$ to participate in the global bandit learning (Line 7). Each sampled client $u \in U_l$ collects their local reward observations of each chosen action $x \in \text{supp}(\pi_l)$ and computes the average $y_l^u(x)$ as feedback (Line 10). Before being used to estimate the global parameter by the central server, these feedback $\vec{y}_l^u \triangleq (y_l^u(x))_{x \in \text{supp}(\pi_l)} \in \mathbb{R}^{|\text{supp}(\pi_l)|}$ are processed by a PRIVATIZER $\mathcal{P}$ to ensure differential privacy. Recall that a PRIVATIZER $\mathcal{P} = (\mathcal{R}, \mathcal{S}, \mathcal{A})$ is a process completed by the clients, the server, and/or a trusted third party. In particular, according to the requirements of privacy protections under different DP models, the PRIVATIZER $\mathcal{P}$ enjoy flexible instantiations (see detailed discussions in Section 4). Generally speaking, a PRIVATIZER works in the following manner: each client $u$ runs the randomizer $\mathcal{R}$ on its local average reward $\vec{y}_l^u$ (over $T_l$ pulls) and then sends the resulting (potentially private) messages $\mathcal{R}(\vec{y}_l^u)$ to the shuffler (Line 13). The shuffler $\mathcal{S}$ (if exists) permutes messages from all clients uniformly before sending the results $\mathcal{S}(\{\mathcal{R}(\vec{y}_l^u)\}_{u \in U_l})$ to the analyzer $\mathcal{A}$ at the central server (Line 15). Finally, the analyzer $\mathcal{A}$

aggregates received messages (potentially in a privacy-preserving manner) and outputs a private averaged local reward $\tilde{y}_l(x)$ (over participating clients $U_l$) for each action $x \in \text{supp}(\pi_l)$ (Line 16). We defer the rigorous formulation of different DP models for the PRIVATIZER $\mathcal{P}$ to Section 4, where the corresponding instantiations of $\mathcal{R}, \mathcal{S}$, and $\mathcal{A}$ are also provided.

**Parameter estimation and action elimination (Lines 17-19):** Using privately aggregated feedback (i.e., the private averaged local reward $\tilde{y}_l$ of the chosen actions $x \in \text{supp}(\pi_l)$), the central server computes the least-square estimator $\tilde{\theta}_l$ (Line 17). To do so, we need to carefully construct a confidence width $W_l$ as follows:

$$W_l \triangleq \left( \underbrace{\sqrt{\frac{2d}{|U_l|h_l}}}_{\text{action-related}} + \underbrace{\frac{\sigma}{\sqrt{|U_l|}}}_{\text{client-related}} + \underbrace{\sigma_n}_{\text{privacy noise}} \right) \sqrt{2 \log \left( \frac{1}{\beta} \right)}, \tag{3}$$

where $\sigma$ is the standard variance associated with client sampling, $\sigma_n$ is related to the privacy noise determined by the DP model, and $\beta$ is the confidence level from the input. We choose this confidence width based on the concentration inequality for sub-Gaussian variables. Specifically, while the first two terms characterize the action-related and client-related uncertainty, respectively, the third term captures the added privacy noise to ensure differential privacy for the clients. Note that this privacy noise $\sigma_n$ depends on the considered DP model. Using this confidence width $W_l$ and the estimated global model parameter $\tilde{\theta}_l$, we can identify a subset of suboptimal actions $E_l$ *w.h.p.* (Line 18). At the end of the $l$-th phase, we update the set of active actions $\mathcal{D}_{l+1}$ by eliminating $E_l$ from $\mathcal{D}_l$ (Line 19).

Finally, we make two remarks about the DP-DPE algorithm.

**Remark 1.** *While a finite number of actions is assumed in this paper, one could extend it to the case with an infinite number of actions by using the covering argument (Lattimore & Szepesvári, 2020, Lemma 20.1). Specifically, when the action set $\mathcal{D} \subseteq \mathbb{R}^d$ is infinite, we can replace $\mathcal{D}$ with a finite set $\mathcal{D}_{\epsilon_0} \subseteq \mathbb{R}^d$ with $|\mathcal{D}_{\epsilon_0}| \le (3/\epsilon_0)^d$ such that for all $x \in \mathcal{D}$, there exists an $x' \in \mathcal{D}_{\epsilon_0}$ with $\|x - x'\|_2 \le \epsilon_0$.*

**Remark 2.** *In Algorithm 1, we assume that $\mathcal{D}_l$ spans $\mathbb{R}^d$ such that matrices $V(\pi_l)$ and $V_l$ are invertible. Then, one could find the near optimal design $\pi_l(\cdot)$ (Line 4) and compute the least-square estimator $\tilde{\theta}_l$ (Line 17). When $\mathcal{D}_l$ does not span $\mathbb{R}^d$, one can simply work in the smaller space $\text{span}(\mathcal{D}_l)$ (Lattimore et al., 2020).*

## 4 DP-DPE in Different DP Models

As alluded before, one of the key features of our general algorithmic framework DP-DPE is that it enables us to consider different trust models in DP (i.e., who the user can trust with her sensitive data) in a unified way by instantiating different mechanisms for the PRIVATIZER. In this section, we formalize DP models integrated with our DP-DLB formulation and provide concrete instantiations for the PRIVATIZER $\mathcal{P}$ in DP-DPE according to different trust models.

### 4.1 DP-DPE in the Central DP Model

In the central DP model, we assume that each client trusts the server, and hence, the server can collect clients' raw data (i.e., the local reward $y_l^u(x)$ for each chosen action $x$ in our case). The privacy guarantee is that any adversary with arbitrary auxiliary information cannot infer a particular client's data by observing the outputs of the server. To achieve this privacy protection, the central DP model requires that the outputs of the server on two neighboring datasets differing in only one client are indistinguishable (Dwork et al., 2006). To present the formal definition in our case, recall that the DP-DPE algorithm (Algorithm 1) runs in phases, and in each phase $l$, a set of new clients $U_l$ will participate in the global bandit learning by providing their feedback. Let[7] $\mathcal{U}_T \triangleq (U_l)_{l=1}^L \in \mathcal{U}^*$ be the sequence of all the participating clients in the total $L$ phases ($T$ rounds). We use $\mathcal{M}(\mathcal{U}_T) = (x_1, \ldots, x_T) \in \mathcal{D}^T$ to denote the sequence of actions chosen in $T$ rounds by the central server. Intuitively, we are interested in a randomized algorithm such that the output $\mathcal{M}(\mathcal{U}_T)$ does not reveal "much" information about any particular client $u \in \mathcal{U}_T$. Formally, we have the following definition.

---

[7] We use the superscript * to indicate that the length could be varying.

**Definition 1.** *(Differential Privacy (DP)). For any $\epsilon \geq 0$ and $\delta \in [0, 1]$, a DP-DPE instantiation is $(\epsilon, \delta)$-differentially private (or $(\epsilon, \delta)$-DP) if for every $\mathcal{U}_T, \mathcal{U}_T' \subseteq \mathcal{U}$ differing on a single client and for any subset of actions $Z \subseteq \mathcal{D}^T$,*

$$\mathbb{P}[\mathcal{M}(\mathcal{U}_T) \in Z] \leq e^\epsilon \mathbb{P}[\mathcal{M}(\mathcal{U}_T') \in Z] + \delta. \tag{4}$$

According to the post-processing property of DP (cf. Proposition 2.1 in Dwork et al. (2014a)), it suffices to guarantee that the final analyzer $\mathcal{A}$ in $\mathcal{P}$ is $(\epsilon, \delta)$-DP. To achieve this, we resort to standard Gaussian mechanism at the server where $\mathcal{A}$ computes the average of local rewards for each chosen action to guarantee $(\epsilon, \delta)$-DP. Specifically, in each phase $l$, the participating clients send their average local rewards $\{\vec{y}_l^u\}_{u \in U_l}$ directly to the central server, and the central server adds Gaussian noise to the average local feedback (over clients) before estimating the global parameter and deciding the chosen actions in the next phase. That is, in the central DP model, both $\mathcal{R}$ and $\mathcal{S}$ of the PRIVATIZER $\mathcal{P}$ are identity mapping while $\mathcal{A}$ adds Gaussian noise when computing the average. In this case, $\mathcal{P} = \mathcal{A}$, and the private aggregated feedback for the chosen actions in the $l$-th phase can be represented as

$$\tilde{y}_l = \mathcal{P}\left(\{\vec{y}_l^u\}_{u \in U_l}\right) = \mathcal{A}\left(\{\vec{y}_l^u\}_{u \in U_l}\right) = \frac{1}{|U_l|} \sum_{u \in U_l} \vec{y}_l^u + (\gamma_1, \ldots, \gamma_{s_l}), \tag{5}$$

where $s_l \triangleq |\mathrm{supp}(\pi_l)|$, $\gamma_j \overset{i.i.d.}{\sim} \mathcal{N}(0, \sigma_{nc}^2)$, and the variance $\sigma_{nc}^2$ is based on the $\ell_2$ sensitivity of the average $\frac{1}{|U_l|} \sum_{u \in U_l} \vec{y}_l^u$. In the rest of the paper, we will continue to use $s_l$ instead of $|\mathrm{supp}(\pi_l)|$ to denote the number of actions chosen in the $l$-th phase for notational simplicity, and it is also the dimension of $\vec{y}_l^u$ for all $u$.

With the above definition, we present the privacy guarantee of DP-DPE in the central DP model in Theorem 1.

**Theorem 1.** *The DP-DPE instantiation using the PRIVATIZER in Eq. 5 with $\sigma_{nc} = \frac{2B\sqrt{2s_l \ln(1.25/\delta)}}{\epsilon |U_l|}$ guarantees $(\epsilon, \delta)$-DP.*

The relatively high trust model in the central DP is not always feasible in practice since some clients do not trust the server and are not willing to share any of their sensitive data. This motivates the introduction of a strictly stronger notion of privacy protection called the local DP (Kasiviswanathan et al., 2011), which is the main focus of the next subsection.

## 4.2 DP-DPE in the Local DP Model

In the local DP model, the privacy burden is now at each client's local side, in the sense that any data sent by any client must already be private. In other words, even though an adversary can observe the data communicated from a client to the server, the adversary cannot infer any sensitive information about the client. Mathematically, this requires a local randomizer $\mathcal{R}$ at each user's side to generate approximately indistinguishable outputs on any two different data inputs. In particular, let $Y$ be the set of all possible values of the average local reward $\vec{y}_l^u$ for client $u$. Then, we have the following formal definition.

**Definition 2.** *(Local Differential Privacy (LDP)). For any $\epsilon \geq 0$ and $\delta \in [0, 1]$, a DP-DPE instantiation is $(\epsilon, \delta)$-local differentially private (or $(\epsilon, \delta)$-LDP) if for any client $u$, every two datasets $\vec{y}, \vec{y}' \in Y$ satisfies*

$$\mathbb{P}[\mathcal{R}(\vec{y}) = o] \leq e^\epsilon \mathbb{P}[\mathcal{R}(\vec{y}') = o] + \delta. \tag{6}$$

*for every possible output $o \in \{\mathcal{R}(\vec{y}) | \vec{y} \in Y\}$.*

That is, an instantiation of DP-DPE is $(\epsilon, \delta)$-LDP if the local randomizer $\mathcal{R}$ in $\mathcal{P}$ is $(\epsilon, \delta)$-DP. To this end, the randomizer $\mathcal{R}$ at each client employs a Gaussian mechanism, the shuffler $\mathcal{S}$ is a simple identity mapping, and the analyzer $\mathcal{A}$ at the server side conducts a simple averaging. Then, the overall output of the PRIVATIZER is the following:

$$\tilde{y}_l = \mathcal{P}\left(\{\vec{y}_l^u\}_{u \in U_l}\right) = \frac{1}{|U_l|} \sum_{u \in U_l} \mathcal{R}(\vec{y}_l^u) = \frac{1}{|U_l|} \sum_{u \in U_l} \left(\vec{y}_l^u + (\gamma_{u,1}, \ldots, \gamma_{u,s_l})\right), \tag{7}$$

where $\gamma_{u,j} \overset{i.i.d.}{\sim} \mathcal{N}(0, \sigma_{nl}^2)$, and the variance $\sigma_{nl}^2$ is based on the sensitivity of $\vec{y}_l^u$.

With the above definition, we present the privacy guarantee of DP-DPE in the local DP model in Theorem 2.

**Theorem 2.** *The DP-DPE instantiation using the PRIVATIZER in Eq. (7) with $\sigma_{nl} = \frac{2B\sqrt{s_l \ln(1.25/\delta)}}{\epsilon}$ guarantees $(\epsilon, \delta)$-LDP.*

Although the local DP model offers a stronger privacy protection compared to the central DP model, it often comes at a price of the regret performance. As we will see, the regret performance of DP-DPE in the local DP model is much worse than that in the central DP model. Therefore, a fundamental question is whether there is a PRIVATIZER for DP-DPE that can achieve the same regret as in the central DP PRIVATIZER while assuming similar trust model as in the local DP PRIVATIZER. This motivates us to consider a recently proposed *shuffle DP model* (Cheu et al., 2019; Erlingsson et al., 2019), which is the main focus of the next subsection.

### 4.3 DP-DPE in the Shuffle DP Model

In the shuffle DP model, between the clients and the server, there exists a shuffler that permutes a batch of clients' randomized data before they are observed by the server so that the server cannot distinguish between two clients' data. Thus, an additional layer of randomness is introduced via shuffling, which can often be easily implemented using cryptographic primitives (e.g., mixnets) due to its simple operation (Bittau et al., 2017). Due to this, the clients now tend to trust the shuffler but still do not trust the central server as in the local DP model. This new trust model offers a possibility to achieve a better regret-privacy tradeoff. This is because the additional randomness of the shuffler creates a *privacy blanket* so that by adding much less random noise, each client can now hide her information in the crowd, i.e., privacy amplification by shuffling (Garcelon et al., 2021).

Formally, a standard one-round shuffle protocol consists of all the three parts: a (local) randomizer $\mathcal{R}$, a shuffler $\mathcal{S}$, and an analyzer $\mathcal{A}$. In this protocol, the clients trust the shuffler but not the analyzer. Hence, the privacy objective is to ensure that the outputs of the shuffler on two neighbouring datasets are indistinguishable from the analyzer's point of view. Note that each client still does not send her raw data to the shuffler even though she trusts it. Due to this, a shuffle protocol often also offers a certain level of LDP guarantee.

In our case, the online learning algorithm will proceed in multiple phases rather than a simple one-round computation. Thus, we need to guarantee that all the shuffled outputs are indistinguishable. To this end, we define the (composite) mechanism $\mathcal{M}_s(\mathcal{U}_T) \triangleq (\mathcal{S} \circ \mathcal{R}^{|U_1|}, \mathcal{S} \circ \mathcal{R}^{|U_2|}, \ldots, \mathcal{S} \circ \mathcal{R}^{|U_L|})$, where $\mathcal{S} \circ \mathcal{R}^{|U_l|} \triangleq \mathcal{S}(\{\mathcal{R}(\vec{y}_l^u)\}_{u \in U_l})$. We say a DP-DPE instantiation satisfies the shuffle differential privacy (SDP) if the composite mechanism $\mathcal{M}_s$ is DP, which leads to the following formal definition.

**Definition 3.** *(Shuffle Differential Privacy (SDP)). For any $\epsilon \geq 0$ and $\delta \in [0, 1]$, a DP-DPE instantiation is $(\epsilon, \delta)$-shuffle differential privacy (or $(\epsilon, \delta)$-SDP) if for any pair $\mathcal{U}_T$ and $\mathcal{U}_T^{'}$ that differ in at most one client, the following is satisfied for all $Z \subseteq Range(\mathcal{M}_s)$:*

$$\mathbb{P}[\mathcal{M}_s(\mathcal{U}_T) \in Z] \leq e^\epsilon \mathbb{P}[\mathcal{M}_s(\mathcal{U}_T^{'}) \in Z] + \delta. \tag{8}$$

We present the concrete pseudocode of $\mathcal{R}$, $\mathcal{S}$, and $\mathcal{A}$ for the shuffle DP model PRIVATIZER $\mathcal{P}$ in Algorithm 2 (see Appendix A.3), which builds on the vector summation protocol recently proposed in Cheu et al. (2021). Here, we provide a brief description of the process. Essentially, the noise added in the shuffle model PRIVATIZER relies on the upper bound of $\ell_2$ norm of the input vectors. However, each component operates on each coordinate of the input vectors independently. Recall that the input of the shuffle model PRIVATIZER is $\{\vec{y}_l^u\}_{u \in U_l}$ and that each chosen action $x$ corresponds to a coordinate in the $s_l$-dimensional vector. Consider the coordinate $j_x$ corresponding to action $x$, and the entry $y_l^u(x)$ at client $u$. First, the local randomizer $\mathcal{R}$ encodes the input $y_l^u(x)$ via a fixed-point encoding scheme (Cheu et al., 2019) and ensures privacy by injecting binomial noise. Specifically, given any scalar $w \in [0, 1]$, it is first encoded as $\hat{w} = \bar{w} + \gamma_1$ using an accuracy parameter $g \in \mathbb{N}$, where $\bar{w} = \lfloor wg \rfloor$ and $\gamma_1 \sim \text{Ber}(wg - \bar{w})$ is a Bernoulli random variable. Then, a binomial noise $\gamma_2 \sim \text{Bin}(b, p)$ is generated, where $b \in \mathbb{N}$ and $p \in (0, 1)$ controls the level of the privacy noise. The output of the local randomizer for each coordinate is simply a collection of $g + b$ bits, where $\hat{w} + \gamma_2$ bits are 1's and the rest are 0's. Combining these $g + b$ bits for each coordinate $j_x$ for $x \in \text{supp}(\pi_l)$ yields the final outputs of the local randomizer $\mathcal{R}$ for the vector $\vec{y}_l^u$. Note that the output bits for each coordinate are marked with the coordinate index so that they will not be mixed up in the following procedures.

After receiving the bits from all participating clients, the shuffler $\mathcal{S}$ simply permutes these bits uniformly at random and sends the output to the analyzer $\mathcal{A}$ at the central server. The analyzer $\mathcal{A}$ adds the received bits, removes the bias introduced by encoding and binormial noise (through simple shifting operations), and divides the result by $|U_l|$ for each coordinate. Finally, the analyzer $\mathcal{A}$ outputs a random $s_l$-dimensional vector $\tilde{y}_l$, whose expectation is the average of the input vectors. That is, $\mathbb{E}[\tilde{y}_l] = \frac{1}{|U_l|} \sum_{u \in U_l} \vec{y}_l^u$ (which is proven in Appendix A.3). In the shuffle model PRIVATIZER, the three parameters $g$, $b$, and $p$ need to be properly chosen according to the privacy requirement. Then, the final privately aggregated data is the following:

$$\tilde{y}_l = \mathcal{P}\left(\{\vec{y}_l^u\}_{u \in U_l}\right) = \mathcal{A}(\mathcal{S}(\{\mathcal{R}(\vec{y}_l^u)\}_{u \in U_l})). \tag{9}$$

With the above definition, we present the privacy guarantee of DP-DPE in the shuffle DP model in Theorem 3.

**Theorem 3.** *For any $\epsilon \in (0, 15)$ and $\delta \in (0, 1/2)$, the DP-DPE instantiation using the PRIVATIZER specified in Algorithm 2 guarantees $(\epsilon, \delta)$-SDP.*

## 5    Performance Analysis

In this section, we study the performance of DP-DPE under different DP models in terms of regret and communication cost. We start with the non-private DP-DPE algorithm (with $\tilde{y}_l = \frac{1}{|U_l|} \sum_{u \in U_l} \vec{y}_l^u$ and $\sigma_n = 0$ for all $l$) and present the main results in Theorem 4.

**Theorem 4** (Non-private DP-DPE). *Suppose $\beta = 1/(kT)$ and $\sigma_n = 0$ in Algorithm 1. Then, the non-private DP-DPE algorithm achieves the following expected regret:*

$$\mathbb{E}[R(T)] = O(\sqrt{dT \log(kT)}) + O(\sigma T^{1-\alpha/2} \sqrt{d \log(kT)}) + O(d \log \log d), \tag{10}$$

*and the communication cost is $O(dT^\alpha)$.*

We present a proof sketch below and provide the detailed proof in Appendix B.

*Proof sketch.* We begin by considering a concentration inequality $P\left\{\langle \tilde{\theta}_l - \theta^*, x \rangle \geq W_l\right\} \leq 2\beta$, which indicates that in the $l$-th phase, the estimation error for the global reward of each action is bound by $W_l$ *w.h.p.* Then, we show that the optimal action stays in the active set the whole time *w.h.p.* and that the regret incurred by one pull is bounded by $4W_{l-1}$ in the $l$-th phase. Finally, summing up the regret over rounds in all phases, we derive the regret upper bound. The analysis of the communication cost is quite straightforward. In the $l$-th phase, only local average reward of each chosen action in this phase is communicated. Since the number of chosen actions is bounded by $(4d \log \log d + 16)$ according to the near-$G$-optimal design (Lattimore et al., 2020, Proposition 3.7), the communication cost is proportional to the total number of clients involved in the entire learning process. $\qquad\square$

**Remark 3.** *Theorem 4 shows an instance-independent regret upper bound for the proposed DP-DPE algorithm. Interestingly, we can observe an obvious tradeoff between regret and communication cost, captured by the value of $\alpha$. While a larger $\alpha$ leads to a smaller regret, it also incurs a larger communication cost. Setting $\alpha = 2/3$ gives $O(T^{2/3})$ for both regret and communication cost.*

In the following, we present the performance of DP-DPE in terms of regret and communication cost under different DP models, i.e., instantiated with different PRIVATIZERs introduced in Section 4. We use CDP-DPE, LDP-DPE, and SDP-DPE to denote the DP-DPE algorithm in the central, local, and shuffle DP models, respectively.

**Theorem 5** (CDP-DPE). *Consider the Gaussian mechanism with $\sigma_{nc} = \frac{2B\sqrt{2s_l \ln(1.25/\delta)}}{\epsilon|U_l|}$ in the central DP model. With $\sigma_n = 2\sigma_{nc}\sqrt{d}$ and $\beta = 1/(kT)$, CDP-DPE achieves the following expected regret:*

$$\mathbb{E}[R(T)] = O(\sigma T^{1-\alpha/2} \sqrt{d \log(kT)}) + O\left(\frac{Bd^{1+\alpha}\sqrt{\ln(1/\delta)}T^{1-\alpha}\sqrt{\log(kT)}}{\epsilon}\right)$$
$$+ O\left(\frac{Bd^{3/2}\sqrt{\ln(1/\delta)}\sqrt{\log(kT)}}{\epsilon}\right), \tag{11}$$

and the communication cost is $O(dT^\alpha)$.

**Theorem 6** (LDP-DPE). *Consider the Gaussian mechanism with $\sigma_{nl} = \frac{2B\sqrt{2s_l \ln(1.25/\delta)}}{\epsilon}$ in the local DP model. With $\sigma_n = 2\sigma_{nl}\sqrt{\frac{d}{|U_l|}}$ in the l-th phase and $\beta = 1/(kT)$, LDP-DPE achieves the following expected regret:*

$$\mathbb{E}[R(T)] = O\left(\left(\frac{Bd\sqrt{\ln(1/\delta)}}{\epsilon} + \sigma\right) T^{1-\alpha/2}\sqrt{d\log(kT)}\right) + O\left(\frac{Bd^2\sqrt{\ln(1/\delta)}\sqrt{\log(kT)}}{\epsilon}\right), \quad (12)$$

*and the communication cost is $O(dT^\alpha)$.*

**Theorem 7** (SDP-DPE). *With $\sigma_n = 2\sigma_{ns}\sqrt{d} = O\left(\frac{B\sqrt{ds_l}\log(s_l/\delta)}{\epsilon|U_l|}\right)$ in the l-th phase and $\beta = 1/(kT)$, SDP-DPE achieves the following expected regret:*

$$\mathbb{E}[R(T)] = O(\sigma T^{1-\alpha/2}\sqrt{d\log(kT)}) + O\left(\frac{Bd^{1+\alpha}\ln(d/\delta)T^{1-\alpha}\sqrt{\log(kT)}}{\epsilon}\right)$$
$$+ O\left(\frac{Bd^{3/2}\ln(d/\delta)\sqrt{\log(kT)}}{\epsilon}\right), \quad (13)$$

*and the communication cost is $O(dT^{3\alpha/2})$.*

We provide the detailed proofs of Theorems 5, 6, and 7 in Appendix B and make the following remarks.

**Remark 4** (Privacy "for-free"). *Comparing the above results with Theorem 4 for the non-private case, we observe that the DP-DPE algorithm enables us to achieve privacy guarantees "for free" in the central and shuffle DP models, in the sense that the additional regret due to privacy protection is only a lower-order additive term. Essentially, this is because the uncertainty introduced by privacy noise is dominated by the client-related uncertainty, which can be captured by our carefully designed confidence width $W_l$ in Eq. (3) and our choice of $\sigma_n$ for different PRIVATIZERs. See more discussions on achieving privacy "for-free" in Section 6.*

**Remark 5** (Regret-privacy tradeoff in the shuffle model). *Consider the regret due to privacy protection. From Theorems 5 and 6, we can see that while the local DP model ensures a stronger privacy guarantee compared to the central DP model, it introduces an additional regret of $O(T^{1-\alpha/2})$ compared to $O(T^{1-\alpha})$ in the central DP model. The shuffle DP model, however, leads to a much better tradeoff between regret and privacy, achieving nearly the same regret guarantee as the central DP model, yet assuming a similar trust model to the local DP model (i.e., without trusting the central server).*

**Remark 6** (Communication cost). *Both CDP-DPE and LDP-DPE consume the same amount of communication resource as the non-private DP-DPE algorithm, counted based on the number of real numbers (Wang et al., 2019). In contrast, SDP-DPE relies only on binary feedback (0/1 bits) from the clients, and thus, the communication cost is counted based on the number of bits. It is worth noting that sending messages consisting of real numbers could be difficult in practice on finite computers (Canonne et al., 2020; Kairouz et al., 2021), and hence in this case it is desirable to adopt SDP-DPE, which incurs a communication cost of $O(dT^{3\alpha/2})$ bits.*

## 6 Discussion on Achieving Privacy "for Free"

Following the remark on privacy "for-free" in the last section, in this section, we draw an interesting connection of our novel bandit online learning problem to private (distributed) supervised learning problems, through which we provide more intuition on why DP-DPE can achieve privacy "for-free". In particular, we compare our problem with differentially private stochastic convex optimization (DP-SCO) (Bassily et al., 2019), where the goal is to approximately minimize the population loss[8] over convex and Lipschitz loss functions given $n$ *i.i.d. $d$*-dimensional samples from a population distribution, while protecting privacy under different trust

---

[8]The population loss for a solution $w$ is given by $\mathcal{L}(w) \triangleq \mathbb{E}_{z \in \mathcal{D}}[l(w, z)]$, where $w$ is the chosen solution (e.g., weights of a classifier), $z$ is a testing sample from the population distribution $\mathcal{D}$, and $l$ is a convex loss function of $w$.

models. More specifically, via noisy stochastic gradient descent (SGD), the excess losses[9] in DP-SCO under various trust models are roughly as follows:

$$\text{Central and Shuffle Model (Bassily et al., 2019; Cheu et al., 2021):} \quad \widetilde{O}\left(\frac{1}{\sqrt{n}} + \frac{\sqrt{d}}{n\epsilon}\right), \quad (14)$$

$$\text{Local Model (Duchi et al., 2018):} \quad \widetilde{O}\left(\frac{1}{\sqrt{n}} + \frac{\sqrt{d}}{\sqrt{n}\epsilon}\right). \quad (15)$$

Recall our main results in Table 1 as follows (ignoring all the logarithmic terms for clarity):

$$\text{Central and Shuffle Model:} \quad \widetilde{O}\left(\sqrt{d}T^{1-\alpha/2} + \frac{d^{1+\alpha}T^{1-\alpha}}{\epsilon}\right), \quad (16)$$

$$\text{Local Model:} \quad \widetilde{O}\left(\sqrt{d}T^{1-\alpha/2} + \frac{d^{3/2}T^{1-\alpha/2}}{\epsilon}\right). \quad (17)$$

Now, one can easily see that in both problems, privacy protection is achieved "for free" in the central and shuffle models, in the sense that the second term (i.e., the additional privacy-dependent term) is a lower-order term compared to the first term in both Eqs. (14) and (16). On the other hand, under the much stronger local model, in both problems, the additional privacy-dependent term is of the same order as the first term in both Eqs. (15) and (17).

We tend to believe that the above interesting connection is not a coincidence. Rather, it provides us with a sharp insight into our introduced DP-DLB formulation. In particular, we know that the first term $1/\sqrt{n}$ in DP-SCO comes from standard concentration results, i.e., how independent samples approximate the true population parameter. Similarly, in our problem, the first term $\sqrt{d}T^{1-\alpha/2}$ comes from the concentration due to client sampling, which is used to approximate the true unknown population parameter $\theta^*$. On the other hand, the second term in DP-SCO is privacy-dependent and comes from the average of noisy gradients. Similarly, in our problem, the second term is due to the average of the local reward vectors with added noise for preserving privacy.

In addition to these useful insights, we believe that this interesting connection also opens the door to a series of important future research directions, in which one can leverage recent advances in DP-SCO to improve our main results (dependence on $d$, communication efficiency, etc.).

## 7 Numerical Results

In this section, we conduct simulations to evaluate the performance of our proposed DP-DPE algorithm, and discuss several interesting observations. For all simulations, we set total rounds $T = 10^6$, population size $|\mathcal{U}| = 10^5$, action dimension $d = 20$, number of actions $k = 10^3$, noise of local models $\sigma = 0.1$, client-sampling parameter $\alpha = 0.8$, confidence probability $\beta = 1/(kT)$ and perform 20 independent runs.

First, we study the regret performance of the DP-DPE algorithm in different DP models. Recall that we use CDP-DPE, LDP-DPE, and SDP-DPE to denote DP-DPE in the central, local, and shuffle DP models, respectively. In Fig. 2(a), we present the final cumulative regret in $T$ rounds of the three algorithms under different values of the privacy budget $\epsilon$. We can observe an obvious tradeoff between the privacy budget and the regret performance for all the DP models: the cumulative regret in $T$ rounds decreases as the privacy requirement becomes less stringent (i.e., a larger $\epsilon$). In addition, it also reflects the regret-privacy tradeoff across different DP models. That is, with the same privacy budget $\epsilon$, while LDP-DPE has the largest regret yet without requiring the clients to trust anyone else (neither the server nor a third party), CDP-DPE achieves the smallest regret but relies on the assumption that the clients trust the server. Interestingly, SDP-DPE achieves a regret fairly close to that of CDP-DPE, yet without the need to trust the server. Along with our

---

[9]The excess loss measures the gap between the chosen solution and the optimal solution in terms of the population loss. That is, the excess loss of $w$ is given by $\mathcal{L}(w) - \min_{w' \in \mathcal{W}} \mathcal{L}(w')$, where $w$ is often the minimizer of the *Empirical Risk Minimization (ERM)* problem: $\hat{\mathcal{L}}(w) \triangleq \frac{1}{n}\sum_{i=1}^{n} l(w, z_i)$, where $\{z_i\}_{i=1}^{n}$ are *i.i.d.* samples from the population distribution. To find the minimizer of ERM, we often resort to SGD.

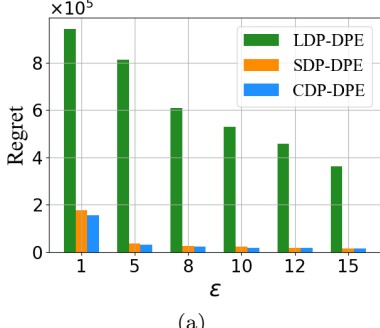 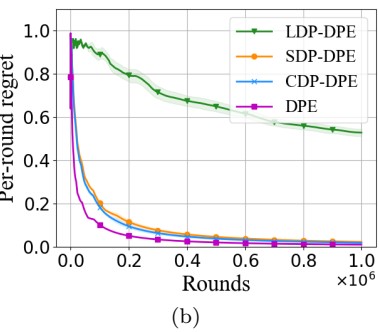 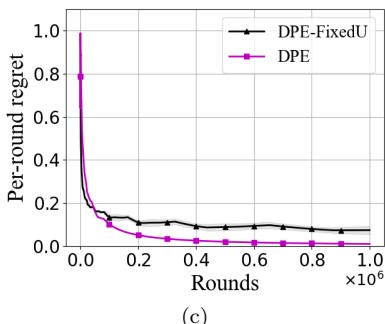

Figure 2: Performance comparisons of different algorithms. The shaded area indicates the standard deviation. (a) Final cumulative regret vs. the privacy budget $\epsilon$. (b) Per-round regret vs. time with privacy parameters $\epsilon = 10$ and $\delta = 0.25$. (c) Per-round regret vs. time for two non-private algorithms.

theoretical results, we demonstrate that DP-DPE in the shuffle model indeed achieves a better regret-privacy tradeoff, compared to the central and local models.

In addition, we are also interested in the regret loss due to privacy protection and how efficiently DP-DPE performs the global bandits learning. Fix the privacy parameters $\epsilon = 10$ and $\delta = 0.25$. In Fig. 2(b), we plot how the per-round regret of the three algorithms (i.e., CDP-DPE, LDP-DPE, and SDP-DPE) varies over time compared to the non-private DP-DPE algorithm, called DPE for simplicity. The DPE algorithm is implemented by letting the PRIVATIZER in DP-DPE simply output the average of the local rewards without adding any privacy noise. We observe that LDP-DPE incurs the largest regret while ensuring the strongest privacy guarantee (i.e., $(\epsilon, \delta)$-LDP). On the other hand, the regret performance of CDP-DPE and SDP-DPE is very close to that of DPE (that does not ensure any privacy guarantees), under the assumption of a trusted central server and a trusted third party shuffler, respectively. Along with our theoretical results, we demonstrate that DP-DPE indeed allows us to achieve privacy "for-free" in the central and shuffle model.

Finally, we demonstrate that the exponentially-increasing client-sampling plays a key role in balancing between regret and communication cost. To this end, we compare our DPE (non-private DP-DPE) algorithm with another non-private algorithm, called DPE-FixedU in Fig. 2(c). The DPE-FixedU algorithm is based on DPE, but differently, samples a fixed number $U$ of participating clients in each phase (i.e., the participating clients are different but the number of clients in each phase is fixed $U$, in contrast to our increasing sampling schedule). For a fair comparison, the value of $U$ is obtained by keeping the total communication costs under two algorithms to be equal. The result shows that DPE learns much faster than the DPE-FixedU when consuming the same communication resource, which validates our conclusion.

## 8  Related Work

The bandit models (including linear bandits) and their variants have proven to be useful for many real-world applications and have been extensively studied (see, e.g., Bubeck & Cesa-Bianchi (2012); Slivkins (2019); Lattimore & Szepesvári (2020) and references therein). Most of the existing studies assume that the exact reward feedback is available to the learning agent for updating the model. However, there is a key difference in the new linear bandit setting we consider: while an action is taken at a central server, it influences a large population of users that contribute to the global reward, which, unfortunately, is not fully observable. Instead, one can learn the global model at the server by randomly sampling a subset of users from the population and iteratively aggregating such partial distributed feedback. While this setting shares some similarities with distributed bandits, federated bandits, and multi-agent cooperative bandits, our motivation and model are very different from theirs, which leads to different regret definitions (global regret vs. group regret; see Section 2) and algorithmic solutions. In the following, we discuss most relevant work in the literature and highlight the key differences.

**Linear bandits.** While the stochastic multi-armed bandits (MAB) model has been extensively studied for a wide range of applications, its modeling power is limited by the assumption that actions are independent. In contrast, the linear bandit model captures the correlation among actions via an unknown parameter (Dani et al., 2008; Rusmevichientong & Tsitsiklis, 2010; Abbasi-Yadkori et al., 2011). The best-known regret upper bound for stochastic linear bandits is $O(d\sqrt{T}\log(T))$ in Abbasi-Yadkori et al. (2011), which holds for an almost arbitrary, even infinite, bounded subset of a finite-dimensional vector space. For a special setting where the set of actions is finite and does not change over time, it is shown in Lattimore & Szepesvári (2020) that a *phased elimination with G-optimal exploration* algorithm guarantees a regret upper bounded by $O(\sqrt{dT\log(kT)})$. This new bound is better by a factor of $\sqrt{d}$, which deserves the effort when $d \geq \log(k)$. However, none of these studies consider the scenario where an action influences a large population and the exact reward feedback is unavailable, which is a key challenge in our problem. Note that the linear bandits we consider is different from the contextual linear bandits in Li et al. (2010); Chu et al. (2011) where the parameter is not shared by actions (although assuming linear reward function), and thus, the actions are not correlated through the parameter.

**Differentially private online learning and bandits.** Since proposed in Dwork et al. (2006), differential privacy (DP) has become the *de facto* privacy preserving model in many applications, including online learning (Jain et al., 2012) and bandits problems (Mishra & Thakurta, 2015). Specifically, in Tossou & Dimitrakakis (2016); Ren et al. (2020); Tenenbaum et al. (2021), MAB has been studied in the central, local, and shuffle DP models, respectively. In Shariff & Sheffet (2018), the authors explore DP in contextual linear bandits and introduce joint DP as ensuring the standard DP incurs a linear regret. As a stronger privacy protection, local DP is also studied for contextual linear bandits (Zheng et al., 2020) and Bayesian optimization (Zhou & Tan, 2020). However, to the best of our knowledge, none of existing works consider the shuffle DP in the linear bandits setting. Moreover, our proposed DP-DPE algorithm allows one to naturally integrate different DP models into a unified algorithmic learning framework.

**Distributed bandits.** Another line of related work is on multi-agent collaborative learning in the distributed bandits setting (Agarwal et al., 2021; Cesa-Bianchi et al., 2016; Martínez-Rubio et al., 2019; Dubey et al., 2020b;a; Wang et al., 2019). The most relevant work to ours is the distributed linear bandit problem studied in Wang et al. (2019). Similarly, they design a distributed phased elimination algorithm where a central server aggregates data provided by the local clients and iteratively eliminates suboptimal actions. However, there are two key differences: i) they consider the standard group regret minimization problem with homogeneous clients that have the same unknown parameter; ii) the observed rewards at the clients are directly sent to the central server without any data privacy protection.

**Federated bandits.** Federated learning (FL) has received substantial attention since its introduction in McMahan et al. (2017). The main idea of FL is to enable collaborative learning among heterogeneous devices while preserving data privacy. Very recently, bandit problems have also been studied in the federated setting, including *federated multi-armed bandits* (Shi & Shen, 2021; Shi et al., 2021; Zhu et al., 2021), *federated linear bandits* (Dubey & Pentland, 2020; Huang et al., 2021), and *federated Bayesian optimization* (Dai et al., 2020). Among all the above work, the two most relevant studies are Huang et al. (2021) and Dubey & Pentland (2020). While they both consider the case where all heterogeneous users share the same unknown parameter with heterogeneous decision sets, in our problem of global reward maximization, the users have heterogeneous unknown local parameters. This heterogeneity among users introduces a new type of uncertainty (called client-related uncertainty), which has to be addressed simultaneously with the action-related uncertainty in our work (see Challenge ⓐ in Section 3.1).

In addition to the differences in model and problem formulation, we also highlight the main technical contributions compared to these works in the following.

Very recently, the work of Huang et al. (2021) studies a similar linear bandits problem in federated setting and also employs a phased elimination algorithm. However, there are two key differences: i) They do not consider the correlation among the actions. That is, the linear bandits setting plays a different role in their work. Specifically, they consider a linear reward for contextual bandits while sill studying multi-armed bandits with independent actions, each of which is associated with a distinct parameter vector. Differently, the linear bandits formulation in our work is used to capture the correlation among the actions and can be extended

to the case with an infinite number of actions; ii) When aggregating users' data for learning the global parameter, we protect users' data privacy using rigorous differential privacy guarantees, which, however, is not considered in their design.

While Dubey & Pentland (2020) also employs DP to protect users' data privacy, in their work both the Gram matrix of actions (of size $O(d^2)$) and reward vectors (of size $O(d)$) need to be periodically communicated using some DP mechanisms (e.g., the Gaussian mechanism). Instead, in our algorithm only private average local reward for the chosen actions (of size $O(d \log \log d)$) would be communicated in each phase. Moreover, while they only consider a variant of the central DP model, our DP-DPE solution provides a unified algorithmic learning framework, which can be instantiated with different DP models. Specifically, DP-DPE with the shuffle model enables us to achieve a finer regret-privacy-communication tradeoff (see Table 1). That is, not only can it achieve nearly the same regret performance as the central model (yet without trusting the central server), but it requires the users to report feedback in bits only throughout the learning process.

**Discussion.** One may wonder whether we can follow the idea of federated learning to share clients' locally learned model parameters only. This way, one can avoid sharing the raw data, which is another way of protecting clients' data privacy. However, we argue that the additional benefit is marginal. On the one hand, by employing different DP mechanisms, our proposed DP-DPE algorithms already ensure provable privacy guarantees. On the other hand, the communication cost of transmitting the (private) average rewards is nearly the same as that of transmitting the local model parameters. Specifically, in each phase, a client in our DP-DPE algorithm needs to send a $|\mathrm{supp}(\pi_l)|$-dimensional vector in DP-DPE, compared to a $d$-dimensional vector when sending the local model parameters. Therefore, the difference is really marginal since we have $|\mathrm{supp}(\pi_l)| \le 4d \log \log d + 16$.

## 9 Conclusion

In this paper, we considered the problem of global reward maximization with only partial feedback in a distributed linear bandit setting. This problem is motivated by several practical applications, where the action of the learning agent would generate a global reward, which represents the overall performance of a large population. In such scenarios, it is often difficult or even impossible to obtain exact feedback of the global reward, which renders existing bandit algorithms inapplicable. To that end, we proposed a novel distributed linear bandits formulation that enables the learning agent to sample clients and interact with them by iteratively aggregating such partial distributed feedback. Two practical challenges arise in this learning process: communication efficiency and data privacy. To address these challenges, we developed a unified algorithmic learning framework, called DP-DPE, which can naturally integrate different differential privacy models and enable us to systematically study the regret-privacy-communication tradeoff. An important takeaway message from our theoretical and simulation results is two-fold: (i) DP-DPE allows us to achieve privacy "for-free" in the central and shuffle models; (ii) DP-DPE in the shuffle model achieves a much better regret-privacy-communication tradeoff compared to the central and local models.

**Future work.** While we assume actions correlated through a common linear reward function with parameter $\theta^*$, one interesting direction for future work is to extend linear reward functions to general functions (possibly non-convex) via kernelized bandits. Another line of interesting direction is to consider misspecified distributed linear bandits where the expected local parameter of the sampled clients is different from the global parameter. In addition, our work also raises several interesting questions that are worth investigating. For example, can we further improve the communication efficiency by using advanced shuffle protocols? Can we generalize our formulation to studying reinforcement learning problems?

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

## A  Proofs of Theorems in Section 4

In this section, we provide the proof of the theorems in Section 4 to show that DP-DPE is differentially private in different models. Before considering different models, we describe the Gaussian Mechanism in the differential privacy literature.

Let $f : \mathcal{U} \to \mathbb{R}^s$ be a vector-valued function operating on databases. The $\ell_2$-sensitivity of $f$, denoted $\Delta_2 f$ is the maximum over all pairs $\mathcal{U}, \mathcal{U}'$ of neighboring datasets of $\|f(\mathcal{U}) - f(\mathcal{U}')\|$. The Gaussian mechanism adds independent noise drawn from a Gaussian with mean zero and standard deviation slightly greater than $\Delta_2 f \sqrt{\ln(1/\delta)}/\epsilon$ to each element of its output (Dwork et al., 2014b).

**Theorem 8.** *(Gaussian Mechanism (Dwork et al., 2014a)). Given any vector-valued function $f : \mathcal{U}^* \to \mathbb{R}^s$, define $\Delta_2 \triangleq \max_{\mathcal{U}, \mathcal{U}' \in \mathcal{U}^*} \|f(\mathcal{U}) - f(\mathcal{U}')\|_2$. Let $\sigma = \Delta_2 \sqrt{2\ln(1.25/\delta)}/\epsilon$. The Gaussian mechanism, which adds independently drawn random noise from $\mathcal{N}(0, \sigma^2)$ to each output of $f(\cdot)$, i.e. returning $f(\mathcal{U}) + (\gamma_1, \ldots, \gamma_s)$ with $\gamma_j \overset{i.i.d.}{\sim} \mathcal{N}(0, \sigma^2)$, ensures $(\epsilon, \delta)$-DP.*

### A.1  The Central Model

The PRIVATIZER $\mathcal{P}$ in the central model adds Gaussian noise to the averaged local performance of each action directly, i.e., at analyzer $\mathcal{A}$ while doing identity mapping with $\mathcal{R}$ and $\mathcal{S}$. That is,

$$\tilde{y}_l = \mathcal{P}\left(\{\vec{y}_l^u\}_{u \in U_l}\right) = \mathcal{A}\left(\{\vec{y}_l^u\}_{u \in U_l}\right) = \frac{1}{|U_l|} \sum_{u \in U_l} \vec{y}_l^u + (\gamma_1, \ldots, \gamma_{s_l}), \tag{18}$$

where $\gamma_j \overset{i.i.d.}{\sim} \mathcal{N}(0, \sigma_{nc}^2)$.

*Proof of Theorem 1.* Recall that $\mathcal{U}_T = (U_l)_{l=1}^L \subseteq \mathcal{U}$ represents the sequence of participating clients in the total $L$ phases. Let $\mathcal{U}_T, \mathcal{U}_T' \subseteq \mathcal{U}$ be the sequence of participating clients differing on a single client and $U_l$, $U_l'$ be the sets of participating clients in $l$-th phase corresponding to $\mathcal{U}_T$ and $\mathcal{U}_T'$ respectively. Note that the $\ell_2$ sensitivity of the average $\frac{1}{|U_l|} \sum_{u \in U_l} \vec{y}_l^u$ is

$$
\begin{aligned}
\Delta_2 &= \max_{\mathcal{U}_T, \mathcal{U}_T'} \left\| \frac{1}{|U_l|} \sum_{u \in U_l} \vec{y}_l^u - \frac{1}{|U_l|} \sum_{u \in U_l'} \vec{y}_l^u \right\|_2 \\
&\leq \frac{1}{|U_l|} \max_{u, u' \in \mathcal{U}} \|\vec{y}_l^u - \vec{y}_l^{u'}\|_2 \\
&\leq \frac{2}{|U_l|} \max_{u \in \mathcal{U}} \|\vec{y}_l^u\|_2 \leq \frac{2B\sqrt{s_l}}{|U_l|}
\end{aligned}
\tag{19}
$$

where the last step is because $s_l$ is the dimension of $y_l^u$ and then $\|\vec{y}_l^u\|_2^2 \leq s_l \|y_l^u(x)\|_1^2 \leq s_l B^2$.

Let $\sigma_{nc} = \frac{2B\sqrt{2s_l \ln(1.25/\delta)}}{\epsilon |U_l|}$. According to Theorem 8, we have that the output $\tilde{y}_l$ of the PRIVATIZER in the central model is $(\epsilon, \delta)$-DP with respect to $\mathcal{U}_T$.

Combining the result of Proposition 2.1 in Dwork et al. (2014a), we derive that DP-DPE with a PRIVATIZER in the central model is $(\epsilon, \delta)$-DP. $\square$

## A.2 The Local Model

The PRIVATIZER $\mathcal{P}$ in the local model applies Gaussian mechanism to the local average performance of each action $(\vec{y}_l^u)$ with $\mathcal{R}$ while doing identity mapping with $\mathcal{S}$ and pure averaging with $\mathcal{A}$. That is,

$$
\tilde{y}_l = \mathcal{P}\left(\{\vec{y}_l^u\}_{u \in U_l}\right) = \frac{1}{|U_l|} \sum_{u \in U_l} \mathcal{R}(\vec{y}_l^u) = \frac{1}{|U_l|} \sum_{u \in U_l} (\vec{y}_l^u + \gamma_u),
$$

where $\gamma_u \overset{i.i.d.}{\sim} \mathcal{N}(0, \sigma_{nl}^2)$.

*Proof of Theorem 2.* An algorithm is $(\epsilon, \delta)$-LDP if the output of the local randomizer $\mathcal{R}$ is $(\epsilon, \delta)$-DP. In the local model of PRIVATIZER, we have

$$
\mathcal{R}(\vec{y}_l^u) = \vec{y}_l^u + \gamma_u.
\tag{20}
$$

For any input of local parameter estimator $\vec{y}_l$, the $\ell_2$ sensitivity is

$$
\Delta_2 = \max_{\vec{y}_l, \vec{y}_l'} \|\vec{y}_l - \vec{y}_l'\|_2 \leq 2 \max \|\vec{y}_l^u\|_2 \leq 2B\sqrt{s_l}.
\tag{21}
$$

Let $\sigma_{nl} = \frac{2B\sqrt{2s_l \ln(1.25/\delta)}}{\epsilon}$. According to Theorem 8, we have that the output of the local randomizer $\mathcal{R}$ of the PRIVATIZER in the local model is $(\epsilon, \delta)$-DP. That is, the DP-DPE algorithm instantiated with the PRIVATIZER in the local model is $(\epsilon, \delta)$-LDP. $\square$

## A.3 The Shuffle Model

In the shuffle model, the PRIVATIZER $\mathcal{P}$ operates under the cooperation of the local randomizer $\mathcal{R}$ at each client, the shuffler $\mathcal{S}$, and the analyzer $\mathcal{A}$ at the central server. First, we present the implementation of each component of the PRIVATIZER $\mathcal{P}$, including $\mathcal{R}$, $\mathcal{S}$, and $\mathcal{A}$, in the shuffle model in Algorithm 2.

The PRIVATIZER $\mathcal{P}$ in the shuffle model adds binary bits to the local average performance of each played action (after being converted to binary representation) at each client, i.e., at local randomizer $\mathcal{R}$, and then shuffles all bits reported from all participating clients via a shuffler $\mathcal{S}$ before sending them to the central server, where the analyzer $\mathcal{A}$ output an unbiased and private estimator of the average of local parameters. That is,

$$
\tilde{y}_l = \mathcal{P}\left(\{\vec{y}_l^u\}_{u \in U_l}\right) = \mathcal{A}(\mathcal{S}(\{\mathcal{R}(\vec{y}_l^u)\}_{u \in U_l})).
\tag{23}
$$

---

**Algorithm 2** $\mathcal{M} : (\epsilon, \delta)$-SDP vector average mechanism for a set $U$ of clients

---

1: **Input:** $\{y_u\}_{u \in U}$, where each $y_u \in \mathbb{R}^s$, $\|y_u\|_2 \leq \Delta_2$

2: Let

$$
\begin{cases}
\hat{\epsilon} = \frac{\epsilon}{18\sqrt{\log(2/\delta)}} \\
g = \max\{\hat{\epsilon}\sqrt{|U|}/(6\sqrt{5\ln((4s)/\delta)}), \sqrt{s}, 10\} \\
b = \lceil \frac{180g^2 \ln(4s/\delta)}{\hat{\epsilon}^2 |U|} \rceil \\
p = \frac{90g^2 \ln(4s/\delta)}{b\hat{\epsilon}^2 |U|}
\end{cases}
\tag{22}
$$

// Local Randomizer
    **function** $\mathcal{R}(y_u)$

3:     **for** coordinate $j \in [s]$ **do**

4:         Shift data to enforce non-negativity: $w_{u,j} = (y_u)_j + \Delta_2, \forall u \in U$

        //randomizer for each entry

5:         Set $\bar{w}_{u,j} \leftarrow \lfloor w_{u,j}g/(2\Delta_2) \rfloor$                         //$\max|(y_u)_j + \Delta_2| \leq 2\Delta_2$

6:         Sample rounding value $\gamma_1 \sim \mathbf{Ber}(w_{u,j}g/(2\Delta_2) - \bar{w}_{u,j})$

7:         Sample privacy noise value $\gamma_2 \sim \mathbf{Bin}(b, p)$

8:         Let $\phi_j^u$ be a multi-set of $(g + b)$ bits associated with the $j$-th coordinate of client $u$, where $\phi_j^u$ consists of $\bar{w}_{u,j} + \gamma_1 + \gamma_2$ copies of 1 and $g + b - (\bar{w}_{i,j} + \gamma_1 + \gamma_2)$ copies of 0

9:     **end for**

10:     Report $\{(j, \phi_j^u)\}_{j \in [s]}$ to the shuffler

    **end function**

// Shuffler
    **function** $\mathcal{S}(\{(j, \vec{\phi}_j)\}_{j \in [s]})$     //$\vec{\phi}_j = (\phi_j^u)_{u \in U}$

11:     **for** each coordinate $j \in [s]$ **do**

12:         Shuffle and output all $(g + b)|U|$ bits in $\vec{\phi}_j$

13:     **end for**

    **end function**

// Analyzer
    **function** $\mathcal{A}(\mathcal{S}(\{(j, \vec{\phi}_j)\}_{j \in [s]})$

14:     **for** coordinate $j \in [s]$ **do**

15:         Compute $z_j \leftarrow \frac{2\Delta_2}{g|U|}((\sum_{i=1}^{(g+b)|U|}(\vec{\phi}_j)_i) - b|U|p)$            // $(\vec{\phi})_i$ denotes the $i$-th bit in $\vec{\phi}_j$

16:         Re-center: $o_j \leftarrow z_j - \Delta_2$

17:     **end for**

18:     Output the estimator of vector average $o = (o_j)_{j \in [s]}$

    **end function**

---

Before proving Theorem 3, we first show that the shuffle protocol in Algorithm 2 is $(\epsilon, \delta)$-SDP.

In this proof, we use $(\cdot)_j$ to denote the $j$-th element of an vector.

**Theorem 9.** *For any $\epsilon \in (0, 15), \delta \in (0, 1)$, Algorithm 2 is $(\epsilon, \delta)$-SDP, unbiased, and has error distribution which is sub-Gaussian with variance $\sigma_{ns}^2 = O\left(\frac{\Delta_2 \ln(s/\delta)}{\epsilon^2 |U|^2}\right)$ and independent of the inputs.*

*Proof.* The proof for the privacy part follows from the SDP guarantee of vector summation protocol in Cheu et al. (2021). In the following, we try to show the remaining part of the above theorem.

Consider an arbitrary coordinate $j \in [s]$. Note that the sum of the messages produced by $\mathcal{R}$ at client $u$ is: $\bar{w}_{u,j} + \gamma_1 + \gamma_2$. Since $\gamma_1$ is drawn from $\mathbf{Ber}(w_{u,j}g/(2\Delta_2) - \bar{w}_{u,j})$, which has expectation $\mathbb{E}[\gamma_1] = w_{u,j}g/(2\Delta_2) - \bar{w}_{u,j}$ and is 1/2-sub-Gaussian according to Hoeffding Lemma. Meanwhile, $\gamma_2 \sim \mathbf{Bin}(b, p)$ indicates $\mathbb{E}[\gamma_2] = bp$ and $\sqrt{b}/2$-sub-Gaussian. Recall that $z_j = \frac{2\Delta_2}{g|U|}((\sum_{i=1}^{(g+b)U}(\vec{\phi}_j)_i) - b|U|p) = \frac{2\Delta_2}{g|U|}(\sum_{u \in U}(\bar{w}_{u,j} + \gamma_1 + \gamma_2) - b|U|p)$

and $o_j = z_j - \Delta_2$. We have

$$
\begin{aligned}
\mathbb{E}[o_j] =& \mathbb{E}[z_j - \Delta_2] = \mathbb{E}\left[\frac{2\Delta_2}{g|U|}\left(\sum_{u \in U}(\bar{w}_{u,j} + \gamma_1 + \gamma_2) - b|U|p\right) - \Delta_2\right] \\
=& \frac{2\Delta_2}{g|U|}\left(\sum_{u \in U}(\bar{w}_{u,j} + \mathbb{E}[\gamma_1] + \mathbb{E}[\gamma_2]) - b|U|p\right) - \Delta_2 \\
=& \frac{2\Delta_2}{g|U|}\left(\sum_{u \in U}\left(\frac{w_{u,j}g}{2\Delta_2} + bp\right) - b|U|p\right) - \Delta_2 = \frac{1}{|U|}\sum_{u \in U}w_{u,j} - \Delta_2 \\
=& \frac{1}{|U|}\sum_{u \in U}((y_u)_j + \Delta_2) - \Delta_2 = \frac{1}{|U|}\sum_{u \in U}(y_u)_j,
\end{aligned}
$$

which indicates the output is unbiased estimator of the average. In addition, according to the property of sub-Gaussian, we have the output $o_j$ satisfies

$$
\begin{aligned}
\mathrm{Var}[o_j] =& \mathrm{Var}[z_j - \Delta_2] \\
=& \mathrm{Var}\left[\frac{2\Delta_2}{g|U|}\left(\sum_{u \in U}(\bar{w}_{u,j} + \gamma_1 + \gamma_2)\right)\right] \\
=& \mathrm{Var}\left[\frac{2\Delta_2}{g|U|}\left(\sum_{u \in U}(\gamma_1 + \gamma_2)\right)\right] \\
\leq& \frac{4\Delta_2^2}{g^2|U|^2}\left(\frac{|U|}{4} + \frac{b|U|}{4}\right) \\
\leq& \frac{\Delta_2^2}{g^2|U|^2}\left(|U| + |U| \cdot (\frac{180g^2\ln(4s/\delta)}{\hat{\epsilon}^2|U|} + 1)\right) \\
=& \frac{\Delta_2^2}{g^2|U|^2}\left(2|U| + \frac{180g^2\ln(4s/\delta)}{\hat{\epsilon}^2}\right) \\
\overset{(a)}{\leq}& \frac{180\Delta_2^2\ln(4s/\delta)}{|U|^2\hat{\epsilon}^2} + \frac{180\Delta_2^2\ln(4s/\delta)}{|U|^2\hat{\epsilon}^2} \\
=& \frac{360\Delta_2^2\ln(4s/\delta)}{|U|^2\hat{\epsilon}^2} = O\left(\frac{\Delta_2^2\ln^2(s/\delta)}{|U|^2\epsilon^2}\right)
\end{aligned}
$$

where $(a)$ is derived from our choice of $g$ in Eq. (22). The output $o_j$ is $O\left(\frac{\Delta_2\ln(s/\delta)}{|U|\epsilon}\right)$-sub-Gaussian. Then, the output vector $o = \{o_j\}_{j \in [s]}$ is a $s$-dimensional $O\left(\frac{\Delta_2\ln(s/\delta)}{|U|\epsilon}\right)$-sub-Gaussian vector according to the definition of the sub-Gaussian vector. $\qquad\square$

Now, we are ready to prove Theorem 3.

*Proof of Theorem 3.* From Theorem 9, we have the shuffle protocol in Algorithm 2 guarantees $(\epsilon, \delta)$-SDP with inputs $\{y_u\}_{u \in U}$. In the DP-DPE algorithm, we apply the shuffle protocol in Algorithm 2 as a PRIVATIZER with inputs $\{\vec{y}_l^u\}_{u \in U_l}$ and $\Delta_2 = \max \|\vec{y}_l^u\|_2 = B\sqrt{s_l}$ in each phase $l$. By admitting new clients in each phase, the DP-DPE algorithm is $(\epsilon, \delta)$-SDP. $\qquad\square$

## B  Proofs of Theorems in Section 5

In Sectoin 5, we present the performance analysis of the DP-DPE algoritm in different DP models. In this section, we provide complete proofs for each of the theorems in Section 5. For reading convenience, we restate all claims before proving them.

## B.1 Proof of Theorem 4

**Theorem 4** (Non-private DP-DPE). *Suppose $\beta = 1/(kT)$ and $\sigma_n = 0$ in Algorithm 1. Then, the non-private DP-DPE algorithm achieves the following expected regret:*

$$\mathbb{E}[R(T)] = O(\sigma T^{1-\alpha/2}\sqrt{d\log(kT)} + \sqrt{Td\log(kT)} + d\log\log d), \tag{24}$$

*and the communication cost is $O(dT^\alpha)$.*

To prove Theorem 4, we start with analyzing regret in a specific phase and then combine all phases together to get the total regret.

**1) Regret in a specific phase**

Let $r_l$ denote the incurred regret in the $l$-th phase, i.e., $r_l \triangleq \sum_{t\in\mathcal{T}_l}\langle\theta^*, x^* - x_t\rangle$.

i) First, the total number of pulls in the $l$-th phase is $T_l$, where $T_l = \sum_{x\in\text{supp}(\pi_l)} T_l(x)$. We have

$$h_l \leq T_l \leq h_l + |\text{supp}(\pi_l)| \leq h_l + h_1,$$

where $|\text{supp}(\pi_l)| \leq 4d\log\log d + 16 = h_1$.

ii) With $\sigma_n = 0$, $W_l = \sqrt{\frac{4d}{|U_l|h_l}\log\left(\frac{1}{\beta}\right)} + \sqrt{\frac{2\sigma^2}{|U_l|}\log\left(\frac{1}{\beta}\right)}$, we have the following concentration inequalities, for any $x \in \mathcal{D}$,

$$P\left\{\langle\tilde{\theta}_l - \theta^*, x\rangle \geq W_l\right\} \leq 2\beta, \quad \text{and} \quad P\left\{\langle\theta^* - \tilde{\theta}_l, x\rangle \geq W_l\right\} \leq 2\beta. \tag{25}$$

*Proof.* We prove the first concentration inequality in Eq. (25) in the following, and the second inequality can be proved symmetrically. Let

$$W_{l,1} \triangleq \sqrt{\frac{4d}{h_l|U_l|}\log\left(\frac{1}{\beta}\right)} \quad \text{and} \quad W_{l,2} \triangleq \sqrt{\frac{2\sigma^2}{|U_l|}\log\left(\frac{1}{\beta}\right)},$$

and thus, $W_l = W_{l,1} + W_{l,2}$. Note that for any action $x \in \mathcal{D}$, the gap between the estimated reward with parameter $\tilde{\theta}_l$ and the true reward with $\theta^*$ satisfies

$$\langle\tilde{\theta}_l - \theta^*, x\rangle = \left\langle\tilde{\theta}_l - \frac{1}{|U_l|}\sum_{u\in U_l}\theta_u + \frac{1}{|U_l|}\sum_{u\in U_l}\theta_u - \theta^*, x\right\rangle = \left\langle\tilde{\theta}_l - \frac{1}{|U_l|}\sum_{u\in U_l}\theta_u, x\right\rangle + \frac{1}{|U_l|}\sum_{u\in U_l}\langle\theta_u - \theta^*, x\rangle.$$

Then, we have

$$\begin{aligned}
&P\left\{\langle\tilde{\theta}_l - \theta^*, x\rangle \geq W_l\right\} \\
=&P\left\{\left\langle\tilde{\theta}_l - \frac{1}{|U_l|}\sum_{u\in U_l}\theta_u, x\right\rangle + \frac{1}{|U_l|}\sum_{u\in U_l}\langle\theta_u - \theta^*, x\rangle \geq W_l\right\} \\
=&P\left\{\left\langle\tilde{\theta}_l - \frac{1}{|U_l|}\sum_{u\in U_l}\theta_u, x\right\rangle + \frac{1}{|U_l|}\sum_{u\in U_l}\langle\theta_u - \theta^*, x\rangle \geq W_{l,1} + W_{l,2}\right\} \\
\leq&P\left\{\left\langle\tilde{\theta}_l - \frac{1}{|U_l|}\sum_{u\in U_l}\theta_u, x\right\rangle \geq W_{l,1}\right\} + P\left\{\frac{1}{|U_l|}\sum_{u\in U_l}\langle\theta_u - \theta^*, x\rangle \geq W_{l,2}\right\}.
\end{aligned} \tag{26}$$

In the following, we try to bound the above two terms, respectively. Under the non-private DP-DPE algorithm, the output of $\mathcal{P}$ is the exact average of local performance, i.e., $\tilde{y}_l = \mathcal{P}\left(\{\tilde{y}_l^u\}_{u\in U_l}\right) = \frac{1}{|U_l|}\sum_{u\in U_l}\tilde{y}_l^u$. Then, the

estimated model parameter satisfies

$$
\begin{aligned}
\tilde{\theta}_l &= V_l^{-1} G_l \\
&= V_l^{-1} \sum_{x \in \text{supp}(\pi_l)} x T_l(x) \tilde{y}_l(x) \\
&= V_l^{-1} \sum_{x \in \text{supp}(\pi_l)} x T_l(x) \frac{1}{|U_l|} \sum_{u \in U_l} y_l^u(x) \\
&= \frac{1}{|U_l|} \sum_{u \in U_l} V_l^{-1} \sum_{x \in \text{supp}(\pi_l)} x T_l(x) y_l^u(x) \\
&= \frac{1}{|U_l|} \sum_{u \in U_l} V_l^{-1} \sum_{x \in \text{supp}(\pi_l)} x \sum_{t \in \mathcal{T}_l(x)} (x^\top \theta_u + \eta_{u,t}) \\
&= \frac{1}{|U_l|} \sum_{u \in U_l} V_l^{-1} \left( \sum_{x \in \text{supp}(\pi_l)} T_l(x) x x^\top \theta_u + \sum_{x \in \text{supp}(\pi_l)} \sum_{t \in \mathcal{T}_l(x)} \eta_{u,t} x \right) \\
&= \frac{1}{|U_l|} \sum_{u \in U_l} V_l^{-1} \left( V_l \theta_u + \sum_{t \in \mathcal{T}_l} \eta_{u,t} x_t \right) \\
&= \frac{1}{|U_l|} \sum_{u \in U_l} \theta_u + \frac{1}{|U_l|} \sum_{u \in U_l} V^{-1} \sum_{t \in \mathcal{T}_l} \eta_{u,t} x_t
\end{aligned}
\tag{27}
$$

For any $x$ in $\mathcal{D}$,

$$
\left\langle x, V_l^{-1} \sum_{t \in \mathcal{T}_l} \eta_{u,t} x_t \right\rangle = \sum_{t \in \mathcal{T}_l} \langle x, V_l^{-1} x_t \rangle \eta_{u,t}.
\tag{28}
$$

Note that $\eta_{u,t}$ is *i.i.d* 1-sub-Gaussian over $t$ and that the chosen action $x_t$ at $t$ is deterministic in the $l$-th phase under the DP-DPE algorithm. Combining the following result,

$$
\sum_{t \in \mathcal{T}_l} \langle x, V_l^{-1} x_t \rangle^2 = x^\top V_l^{-1} \left( \sum_{t \in \mathcal{T}_l} x_t x_t^\top \right) V_l^{-1} x = \|x\|_{V_l^{-1}}^2,
$$

where the second equality is due to $V_l = \sum_{t \in \mathcal{T}_l} x_t x_t^\top$, we derive that the LHS of Eq. (28) is $\|x\|_{V_l^{-1}}$-sub-Gaussian. Besides, we have $\|x\|_{V_l^{-1}}^2 \le \frac{\|x\|_{V_l(\pi_l)^{-1}}^2}{h_l} \le \frac{g(\pi_l)}{h_l} \le \frac{2d}{h_l}$ by the near-G-optimal design. According to the property of a sub-Gaussian random variable, we can obtain

$$
P\left\{ \frac{1}{|U_l|} \sum_{u \in U_l} \left\langle x, V_l^{-1} \sum_{t \in \mathcal{T}_l} \eta_{u,t} x_t \right\rangle \ge W_{l,1} \right\} \le \exp\left\{ -\frac{|U_l| W_{l,1}^2}{2\|x\|_{V_l^{-1}}^2} \right\} \le \exp\left\{ -\frac{|U_l| \frac{4d}{h_l |U_l|} \log(1/\beta)}{2 \cdot \frac{2d}{h_l}} \right\} = \beta.
\tag{29}
$$

Combining the result in Eq. (27), we have

$$
P\left\{ \left\langle \tilde{\theta}_l - \frac{1}{|U_l|} \sum_{u \in U_l} \theta_u, x \right\rangle \ge W_{l,1} \right\} = P\left\{ \frac{1}{|U_l|} \sum_{u \in U_l} \left\langle x, V_l^{-1} \sum_{t \in \mathcal{T}_l} \eta_{u,t} x_t \right\rangle \ge W_{l,1} \right\} \le \beta
$$

For the second term of Eq. (26), we know that $\langle \theta_u - \theta^*, x \rangle = \langle \xi_u, x \rangle$ is $\|x\|_2 \sigma$-sub-Gaussian. Similarly, according to the sub-Gaussian property, we have

$$
P\left\{ \frac{1}{|U_l|} \sum_{u \in U_l} \langle \theta_u - \theta^*, x \rangle \ge W_{l,2} \right\} \le \exp\left\{ -\frac{|U_l| W_{l,2}^2}{2\|x\|_2^2 \sigma^2} \right\} \le \exp\left\{ -\frac{|U_l| \cdot \frac{2\sigma^2}{|U_l|} \log(\frac{1}{\beta})}{2\sigma^2} \right\} = \beta.
\tag{30}
$$

Therefore, we have

$$P\left\{\langle \tilde{\theta}_l - \theta^*, x \rangle \geq W_l \right\} \leq 2\beta.$$

The symmetrical argument completes the proof. □

iii) Recall that $x^* = \arg\max_{x \in \mathcal{D}} \langle \theta^*, x \rangle$ is the unique optimal action in $\mathcal{D}$. Under DP-DPE algorithm, we define a "good" event at $l$-th phase as $\mathcal{E}_l$:

$$\mathcal{E}_l \triangleq \left\{ \langle \theta^* - \tilde{\theta}_l, x^* \rangle \leq W_l \quad \text{and} \quad \forall x \in \mathcal{D} \backslash \{x^*\} \quad \langle \tilde{\theta}_l - \theta^*, x \rangle \leq W_l \right\}.$$

It is not difficult to derive $P(\mathcal{E}_l) \geq 1 - 2k\beta$ via union bound.

Under event $\mathcal{E}_l$, we have the following two observations:

1. If the optimal action $x^* \in \mathcal{D}_l$, then $x^* \in \mathcal{D}_{l+1}$.

2. For any $x \in \mathcal{D}_{l+1}$, we have $\langle \theta^*, x^* - x \rangle \leq 4W_l$.

Then, for any $l \geq 2$, with probability at least $1 - 2k\beta$, the regret in $l$-th phase $r_l$ satisfies:

$$r_l \leq 8\sqrt{2dh_1 \log(1/\beta)} \left( \sqrt{2^{l-1}} + \frac{1}{\sqrt{2^{l-1}}} \right) + 4h_1\sigma\sqrt{2\log(1/\beta)} \left( \sqrt{2^{(2-\alpha)(l-1)}} + \frac{1}{\sqrt{2^{\alpha(l-1)}}} \right). \tag{31}$$

*Proof.* We first prove the two observations under event $\mathcal{E}_l$ and then show that the regret in the $l$-th phase satisfies the above result.

**Observation 1:**

Let $b \in \arg\max_{x \in \mathcal{D}_l} \langle \tilde{\theta}_l, x \rangle$. If $x^* = b$, then $x^* \in \mathcal{D}_{l+1}$ according to the elimination step in Algorithm 1. If $x^* \neq b$, then under event $\mathcal{E}_l$, we have

$$\begin{aligned}
\langle \tilde{\theta}_l, b - x^* \rangle &= \langle \tilde{\theta}_l, b \rangle - \langle \tilde{\theta}_l, x^* \rangle \\
&\leq \langle \theta^*, b \rangle + W_l - \langle \theta^*, x^* \rangle + W_l \\
&= \langle \theta^*, b - x^* \rangle + 2W_l \\
&\leq 2W_l,
\end{aligned} \tag{32}$$

which means that $x^*$ is not eliminated at the end of the $l$-th phase, i.e., $x^* \in \mathcal{D}_{l+1}$.

**Observation 2:**

For any $x \in \mathcal{D}_{l+1}$, we have $\langle \tilde{\theta}_l, b - x \rangle \leq 2W_l$. Then, we have the following steps:

$$\begin{aligned}
2W_l &\geq \langle \tilde{\theta}_l, b - x \rangle \\
&\geq \langle \tilde{\theta}_l, x^* - x \rangle \\
&\geq \langle \theta^*, x^* \rangle - W_l - \langle \theta^*, x \rangle - W_l \\
&= \langle \theta^*, x^* - x \rangle - 2W_l,
\end{aligned} \tag{33}$$

where the second inequality is from Observation 1. Then, we derive Observation 2.

**Bound regret $r_l$:**

Note that $h_l = 2^{l-1}h_1$ and $2^{\alpha l} \leq |U_l| \leq 2^{\alpha l} + 1$. We have

1) $W_{l,1} = \sqrt{\frac{4d}{h_l|U_l|} \log\left(\frac{1}{\beta}\right)} \leq \sqrt{\frac{4d}{2^{(1+\alpha)l-1}h_1} \log\left(\frac{1}{\beta}\right)}.$

2) $W_{l,2} = \sqrt{\frac{2\sigma^2}{|U_l|} \log\left(\frac{1}{\beta}\right)} \leq \sqrt{\frac{2\sigma^2}{2^{\alpha l}} \log\left(\frac{1}{\beta}\right)}.$

Combining the upper bound of $T_l$, we derive the regret in any phase $l \geq 2$,

$$
\begin{aligned}
r_l &= \sum_{t \in \mathcal{T}_l} \langle \theta^*, x^* - x_t \rangle \\
&\leq \sum_{t \in \mathcal{T}_l} 4W_{l-1} \\
&= \sum_{t \in \mathcal{T}_l} 4(W_{l-1,1} + W_{l-1,2}) \\
&= 4T_l(W_{l-1,1} + W_{l-1,2}) \\
&\leq \underbrace{4h_1(2^{l-1}+1)\sqrt{\frac{4d}{2^{(1+\alpha)(l-1)-1}h_1}\log\left(\frac{1}{\beta}\right)}}_{\text{①}} + \underbrace{4h_1(2^{l-1}+1)\sqrt{\frac{2\sigma^2}{2^{\alpha(l-1)}}\log\left(\frac{1}{\beta}\right)}}_{\text{②}}.
\end{aligned}
\tag{34}
$$

We derive an upper bound for each of the two terms in the above equation.

For ①, we have

$$
\begin{aligned}
\text{①} &\leq 4h_1(2^{l-1}+1)\sqrt{\frac{8d}{2^{(1+\alpha)(l-1)}h_1}\log\left(\frac{1}{\beta}\right)} \\
&= 8\sqrt{2dh_1\log\left(\frac{1}{\beta}\right)}\left(\sqrt{2^{(1-\alpha)(l-1)}} + \frac{1}{\sqrt{2^{(\alpha+1)(l-1)}}}\right) \\
&\leq 8\sqrt{2dh_1\log\left(\frac{1}{\beta}\right)}\left(\sqrt{2^{l-1}} + \frac{1}{\sqrt{2^{l-1}}}\right).
\end{aligned}
\tag{35}
$$

As to the second term ②, we have

$$
\begin{aligned}
\text{②} &\leq 4h_1(2^{l-1}+1)\sqrt{\frac{2\sigma^2}{2^{\alpha(l-1)}}\log\left(\frac{1}{\beta}\right)} \\
&= 4h_1\sigma\sqrt{2\log\left(\frac{1}{\beta}\right)}\left(\sqrt{2^{(2-\alpha)(l-1)}} + \frac{1}{\sqrt{2^{\alpha(l-1)}}}\right).
\end{aligned}
\tag{36}
$$

Then, the regret in the $l$-th phase $r_l$ is upper bounded by

$$
r_l \leq \text{①} + \text{②}.
$$

$\qquad\qquad\qquad\qquad\qquad\qquad\qquad\qquad\qquad\qquad\qquad\qquad\qquad\qquad\qquad\qquad\qquad\qquad\square$

**2) Total Regret**

The expected total regret in $T$ rounds under the non-private DP-DPE algorithm satisfies

$$
\mathbb{E}[R(T)] = O(T^{1-\alpha/2}\sqrt{d\log(kT)} + d\log\log d).
$$

*Proof.* Define $\mathcal{E}_g$ as the event where the "good" event occurs in every phase, i.e., $\mathcal{E}_g \triangleq \bigcap_{l=1}^{L} \mathcal{E}_l$. It is not difficult to obtain $P\{\mathcal{E}_g\} \geq 1 - 2k\beta L$ by applying union bound. At the same time, let $R_g$ be the regret under event $\mathcal{E}_g$, and $R_b$ be the regret if event $\mathcal{E}_g$ does not hold. Then, the expected total regret in $T$ is $\mathbb{E}[R(T)] = P(\mathcal{E}_g)R_g + (1 - P(\mathcal{E}_g))R_b$.

Under event $\mathcal{E}_g$, the regret in the $l$-th phase $r_l$ satisfies Eq. (31) for any $l \geq 2$. Combining $r_1 \leq 2T_1 \leq 4h_1$ ( since $\langle \theta^*, x^* - x \rangle \leq 2$ for all $x \in \mathcal{D}$), we have

$$
\begin{aligned}
R_g &= \sum_{l=1}^{L} r_l \\
&\leq 4h_1 + \sum_{l=2}^{L} 8\sqrt{2dh_1 \log(1/\beta)} \left( \sqrt{2^{l-1}} + \frac{1}{\sqrt{2^{l-1}}} \right) \\
&\quad + \sum_{l=2}^{L} 4h_1 \sigma \sqrt{2\log(1/\beta)} \left( \sqrt{2^{(2-\alpha)(l-1)}} + \frac{1}{\sqrt{2^{\alpha(l-1)}}} \right) \\
&\leq 4h_1 + 8\sqrt{2dh_1 \log(1/\beta)} \cdot 4\sqrt{2^{L-1}} \\
&\quad + 4h_1 \sigma \sqrt{2\log(1/\beta)} \left( \frac{\sqrt{2^{2-\alpha}}}{\sqrt{2^{2-\alpha}} - 1} \cdot \sqrt{2^{(L-1)(2-\alpha)}} + C_1 \right) \\
&= 4h_1 + 8\sqrt{2dh_1 \log(1/\beta)} \cdot 4\sqrt{2^{L-1}} \\
&\quad + 4h_1 \sigma \sqrt{2\log(1/\beta)} \left( 4\sqrt{2^{(L-1)(2-\alpha)}} + C_1 \right),
\end{aligned}
\tag{37}
$$

where $C_1 = \sum_{l=2}^{\infty} \frac{1}{\sqrt{2^{\alpha(l-1)}}}$. Note that $h_L \leq T_L \leq T$, which indicates $2^{L-1} \leq T/h_1$, and $L \leq \log(2T/h_1)$. Then, the above inequality becomes

$$
\begin{aligned}
R_g &= \sum_{l=1}^{L} r_l \\
&\leq 4h_1 + 8\sqrt{2dh_1 \log(1/\beta)} \cdot 4\sqrt{T/h_1} + 4h_1 \sigma \sqrt{2\log(1/\beta)} \left( 4\sqrt{(T/h_1)^{2-\alpha}} + C_1 \right) \\
&\leq 4h_l + 32\sqrt{2dT \log(1/\beta)} + 16\sigma \sqrt{2h_1 \log(1/\beta)} \cdot T^{1-\alpha/2} + 4C_1 h_1 \sigma \sqrt{2\log(1/\beta)}.
\end{aligned}
\tag{38}
$$

On the other hand, $R_b \leq 2T$ since $\langle \theta^*, x^* - x \rangle \leq 2$ for all $x \in \mathcal{D}$. Choose $\beta = \frac{1}{kT}$ in Algorithm 1. Finally, we have the following results:

$$
\begin{aligned}
\mathbb{E}[R(T)] &= P(\mathcal{E}_g) R_g + (1 - P(\mathcal{E}_g)) R_b \\
&\leq R_g + 2k\beta L \cdot 2T \\
&\leq 4h_1 + 32\sqrt{2dT \log(kT)} + 16\sigma \sqrt{2h_1 \log(kT)} \cdot T^{1-\alpha/2} + 4C_1 h_1 \sigma \sqrt{2\log(1/\beta)} + 4\log(2T/h_1) \\
&= O(\sqrt{dT \log(kT)}) + O(\sigma T^{1-\alpha/2} \sqrt{d\log(kT)}) + O(d\log\log d).
\end{aligned}
\tag{39}
$$

$\square$

**3) Communication cost.** Notice that the communicating data in each phase is the local average performance $y_l^u(x)$ for each chosen action $x$ in the support set $\text{supp}(\pi_l)$. Therefore, the total communication cost is

$$
C(T) = \sum_{l=1}^{L} s_l |U_l| \leq \sum_{l=1}^{L} (4d\log\log d + 16) \cdot 2^{\alpha l} = O(dT^\alpha).
\tag{40}
$$

## B.2 Proof of Theorem 5

**Theorem 5** (CDP-DPE). *Consider the Gaussian mechanism with $\sigma_{nc} = \frac{2B\sqrt{2s_l \ln(1.25/\delta)}}{\epsilon|U_l|}$ in the central DP model. With $\sigma_n = 2\sigma_{nc}\sqrt{d}$ and $\beta = 1/(kT)$, CDP-DPE achieves the following expected regret:*

$$\mathbb{E}[R(T)] = O(\sigma T^{1-\alpha/2}\sqrt{d\log(kT)})$$
$$+ O\left(\frac{Bd^{1+\alpha}\sqrt{\ln(1/\delta)}T^{1-\alpha}\sqrt{\log(kT)}}{\epsilon}\right) \tag{41}$$
$$+ O\left(\frac{Bd^{3/2}\sqrt{\ln(1/\delta)}\sqrt{\log(kT)}}{\epsilon}\right)$$

*and the communication cost is $O(dT^\alpha)$.*

Following a similar line to the proof for Theorem 4, we start with analyzing regret in a specific phase and then combine all phases together to get the total regret.

**1) Regret in a specific phase**

Let $r_l$ denote the incurred regret in the $l$-th phase, i.e., $r_l \triangleq \sum_{t\in\mathcal{T}_l}\langle\theta^*, x^* - x_t\rangle$.

i) First, the total number of pulls in the $l$-th phase is $T_l$, where $T_l = \sum_{x\in\mathrm{supp}(\pi_l)} T_l(x)$. We have

$$h_l \leq T_l \leq h_l + |\mathrm{supp}(\pi_l)| \leq h_l + h_1,$$

where $|\mathrm{supp}(\pi_l)| \leq 4d\log\log d + 16 = h_1$.

ii) With $\sigma_n = 2\sigma_{nc}\sqrt{d}$, we have $W_l = \sqrt{\frac{4d}{|U_l|h_l}\log\left(\frac{1}{\beta}\right)} + \sqrt{\frac{2\sigma^2}{|U_l|}\log\left(\frac{1}{\beta}\right)} + \sqrt{8\sigma_{nc}^2 d\log\left(\frac{1}{\beta}\right)}$. We have the following concentration inequalities, for any $x \in \mathcal{D}$,

$$P\left\{\langle\tilde{\theta}_l - \theta^*, x\rangle \geq W_l\right\} \leq 3\beta, \tag{42}$$

and

$$P\left\{\langle\theta^* - \tilde{\theta}_l, x\rangle \geq W_l\right\} \leq 3\beta. \tag{43}$$

*Proof.* We prove the first concentration inequality in Eq. (42) in the following, and the second inequality can be proved symmetrically. Note that $W_l = W_{l,1} + W_{l,2} + W_{l,3}$, where

$$W_{l,1} = \sqrt{\frac{4d}{h_l|U_l|}\log\left(\frac{1}{\beta}\right)}+, \quad W_{l,2} = \sqrt{\frac{2\sigma^2}{|U_l|}\log\left(\frac{1}{\beta}\right)}, \quad \text{and} \quad W_{l,3} \triangleq \sqrt{8d\sigma_{nc}^2\log\left(\frac{1}{\beta}\right)}.$$

Under the DP-DPE algorithm with the central model PRIVATIZER, the output of the PRIVATIZER $\mathcal{P}$ is, $\tilde{y}_l = \frac{1}{|U_l|}\sum_{u\in U_l}\vec{y}_l^u + (\gamma_1, \ldots, \gamma_{s_l})$, where $\gamma_j \overset{i.i.d.}{\sim} \mathcal{N}(0, \sigma_{nc}^2)$. Let $j_x$ denote the index corresponds to the action $x$ in the support set $\mathrm{supp}(\pi_l)$, i.e., $\tilde{y}_l(x) = \frac{1}{|U_l|}\sum_{u\in U_l} y_l^u(x) + \gamma_{j_x}$. Then, the estimated model parameter satisfies

$$\tilde{\theta}_l = V_l^{-1}G_l$$
$$= V_l^{-1}\sum_{x\in\mathrm{supp}(\pi_l)} xT_l(x)\tilde{y}_l(x)$$
$$= V_l^{-1}\sum_{x\in\mathrm{supp}(\pi_l)} xT_l(x)\left(\frac{1}{|U_l|}\sum_{u\in U_l} y_l^u(x) + \gamma_{j_x}\right) \tag{44}$$
$$= \frac{1}{|U_l|}\sum_{u\in U_l} V_l^{-1}\sum_{x\in\mathrm{supp}(\pi_l)} xT_l(x)y_l^u(x) + V_l^{-1}\sum_{x\in\mathrm{supp}(\pi_l)} xT_l(x)\gamma_{j_x}$$
$$= \frac{1}{|U_l|}\sum_{u\in U_l}\theta_u + \frac{1}{|U_l|}\sum_{u\in U_l} V^{-1}\sum_{t\in\mathcal{T}_l}\eta_{u,t}x_t + V_l^{-1}\sum_{x\in\mathrm{supp}(\pi_l)} xT_l(x)\gamma_{j_x}$$

For any action $x' \in \mathcal{D}$, the gap between the estimated reward with parameter $\tilde{\theta}_l$ and the true reward with $\theta^*$ satisfies

$$\langle \tilde{\theta}_l - \theta^*, x' \rangle$$

$$= \left\langle \tilde{\theta}_l - \frac{1}{|U_l|} \sum_{u \in U_l} \theta_u + \frac{1}{|U_l|} \sum_{u \in U_l} \theta_u - \theta^*, x' \right\rangle$$

$$= \left\langle \tilde{\theta}_l - \frac{1}{|U_l|} \sum_{u \in U_l} \theta_u, x' \right\rangle + \frac{1}{|U_l|} \sum_{u \in U_l} \langle \theta_u - \theta^*, x' \rangle$$

$$= \frac{1}{|U_l|} \sum_{u \in U_l} \left\langle V_l^{-1} \sum_{t \in \mathcal{T}_l} \eta_{u,t} x_t, x' \right\rangle + \left\langle V_l^{-1} \sum_{x \in \text{supp}(\pi_l)} x T_l(x) \gamma_{j_x}, x' \right\rangle + \frac{1}{|U_l|} \sum_{u \in U_l} \langle \theta_u - \theta^*, x' \rangle,$$

Then, we have

$$P\left\{ \langle \tilde{\theta}_l - \theta^*, x \rangle \geq W_l \right\}$$

$$= P\left\{ \frac{1}{|U_l|} \sum_{u \in U_l} \left\langle V_l^{-1} \sum_{t \in \mathcal{T}_l} \eta_{u,t} x_t, x' \right\rangle + \left\langle V_l^{-1} \sum_{x \in \text{supp}(\pi_l)} x T_l(x) \gamma_{j_x}, x' \right\rangle + \frac{1}{|U_l|} \sum_{u \in U_l} \langle \theta_u - \theta^*, x' \rangle \geq W_l \right\}$$

$$\leq P\left\{ \frac{1}{|U_l|} \sum_{u \in U_l} \left\langle x', V_l^{-1} \sum_{t \in \mathcal{T}_l} \eta_{u,t} x_t \right\rangle \geq W_{l,1} \right\} + P\left\{ \frac{1}{|U_l|} \sum_{u \in U_l} \langle \theta_u - \theta^*, x \rangle \geq W_{l,2} \right\} \qquad (45)$$

$$+ P\left\{ \left\langle x', V_l^{-1} \sum_{x \in \text{supp}(\pi_l)} x T_l(x) \gamma_{j_x} \right\rangle \geq W_{l,3} \right\}.$$

We have shown that the first and the second probability in the above equation is shown to be less than $\beta$ in Eq. (29) and Eq. (30), respectively. In the following, we try to show that the third term is less than $\beta$.

Note that,

$$\left\langle x', V_l^{-1} \sum_{x \in \text{supp}(\pi_l)} x T_l(x) \gamma_{j_x} \right\rangle = \sum_{x \in \text{supp}(\pi_l)} \langle x', V_l^{-1} x \rangle T_l(x) \gamma_{j_x}, \qquad (46)$$

and that $\gamma_j \overset{i.i.d.}{\sim} \mathcal{N}(0, \sigma_{nc}^2)$. The variance (denoted by $\sigma_{\text{sum}}^2$) of the above sum of $i.i.d.$ Gaussian variables is

$$\sigma_{\text{sum}}^s = \sum_{x \in \text{supp}(\pi_l)} \langle x', V_l^{-1} x \rangle^2 T_l(x)^2 \sigma_{nc}^2 \overset{(a)}{\leq} T_l \cdot x'^\top V_l^{-1} \left( \sum_{x \in \text{supp}(\pi_l)} T_l(x) x x^\top \right) V_l^{-1} x' \sigma_{nc}^2 = T_l \|x'\|_{V_l^{-1}}^2 \sigma_{nc}^2,$$

where $(a)$ is from $T_l(x) \leq T_l$ for any $x$ in the support set $\text{supp}(\pi_l)$. Therefore, the LHS of Eq. (46) is a Gaussian variable with variance

$$\sigma_{\text{sum}}^2 \leq T_l \|x'\|_{V_l^{-1}}^2 \sigma_{nc}^2 \leq T_l \cdot \frac{2d}{h_l} \cdot \sigma_{nc}^2 \overset{(a)}{\leq} 4 d \sigma_{nc}^2,$$

where $(a)$ is due to $T_l \leq h_l + h_1 \leq 2h_l$. Combining the tail bound for Gaussian variable, we have

$$P\left\{ \left\langle x', V_l^{-1} \sum_{x \in \text{supp}(\pi_l)} x T_l(x) \gamma_{j_x} \right\rangle \geq W_{l,3} \right\} \leq \exp\left\{ -\frac{W_{l,3}^2}{2\sigma_{\text{sum}}^2} \right\} \leq \exp\left\{ -\frac{8 d \sigma_{nc}^2 \log(1/\beta)}{8 d \sigma_{nc}^2} \right\} = \beta$$

By now, we complete the proof for Eq. (42). With the symmetrical argument, we obtain the result in Eq. (43). $\qquad \square$

iii) Define event $\mathcal{E}_l$ in the $l$-th phase as follows:

$$\mathcal{E}_l \triangleq \left\{ \langle \theta^* - \tilde{\theta}_l, x^* \rangle \leq W_l \quad \text{and} \quad \forall x \in \mathcal{D} \backslash \{x^*\} \quad \langle \tilde{\theta}_l - \theta^*, x \rangle \leq W_l \right\}.$$

It is not difficult to derive $P(\mathcal{E}_l) \geq 1 - 3k\beta$ via union bound.

Then, our second step is to show that under event $\mathcal{E}_l$ for any $l \geq 2$, the regret $r_l \triangleq \sum_{t \in \mathcal{T}_l} \langle \theta^*, x^* - x_t \rangle$ incurred in $l$-th phase satisfies

$$
\begin{aligned}
r_l \leq & 8\sqrt{2dh_1 \log(1/\beta)} \left( \sqrt{2^{l-1}} + \frac{1}{\sqrt{2^{l-1}}} \right) \\
& + 4h_1\sigma\sqrt{2\log(1/\beta)} \left( \sqrt{2^{(2-\alpha)(l-1)}} + \frac{1}{\sqrt{2^{\alpha(l-1)}}} \right) \\
& + 8h_1\sigma_0\sqrt{2dh_1 \log\left(\frac{1}{\beta}\right)} \left( 2^{(1-\alpha)(l-1)} + \frac{1}{2^{\alpha(l-1)}} \right),
\end{aligned}
\tag{47}
$$

where $\sigma_0 = \frac{2B\sqrt{2\ln(1.25/\delta)}}{\epsilon}$.

*Proof.* To get the bound in Eq. (47) , we start with two observations under event $\mathcal{E}_l$:

1. If the optimal action $x^* \in \mathcal{D}_l$, then $x^* \in \mathcal{D}_{l+1}$.

2. For any $x \in \mathcal{D}_{l+1}$, we have $\langle \theta^*, x^* - x \rangle \leq 4W_l$.

**Bound regret $r_l$:**

Note that $h_l = 2^{l-1}h_1$ and $2^{\alpha l} \leq |U_l| \leq 2^{\alpha l} + 1$. We have

1) $W_{l,1} = \sqrt{\frac{4d}{h_l|U_l|} \log\left(\frac{1}{\beta}\right)} \leq \sqrt{\frac{4d}{2^{(1+\alpha)l-1}h_1} \log\left(\frac{1}{\beta}\right)}$;

2) $W_{l,2} = \sqrt{\frac{2\sigma^2}{|U_l|} \log\left(\frac{1}{\beta}\right)} \leq \sqrt{\frac{2\sigma^2}{2^{\alpha l}} \log\left(\frac{1}{\beta}\right)}$;

3) $W_{l,3} = \sqrt{8d\sigma_{nc}^2 \log\left(\frac{1}{\beta}\right)} = \sqrt{\frac{8ds_l\sigma_0^2}{|U_l|^2} \log\left(\frac{1}{\beta}\right)} \leq \sqrt{\frac{8ds_l\sigma_0^2}{2^{2\alpha l}} \log\left(\frac{1}{\beta}\right)}$, where $\sigma_0 = \frac{2B\sqrt{2\ln(1.25/\delta)}}{\epsilon}$.

Combining the upper bound of $T_l$ (i.e., $T_l \leq h_l + h_1$), we derive the regret in any phase $l \geq 2$,

$$r_l = \sum_{t=t_l}^{t_l+T_l-1} \langle \theta^*, x^* - x_t \rangle$$

$$\leq \sum_{t=t_l}^{t_l+T_l-1} 4W_{l-1}$$

$$= \sum_{t=t_l}^{t_l+T_l-1} 4(W_{l-1,1} + W_{l-1,2} + W_{l-1,3})$$

$$= 4T_l(W_{l-1,1} + W_{l-1,2} + W_{l-1,3})$$

$$\leq \underbrace{4h_1(2^{l-1}+1)\sqrt{\frac{4d}{2^{(1+\alpha)(l-1)-1}h_1}\log\left(\frac{1}{\beta}+\right)}}_{\text{①}} \tag{48}$$

$$+ \underbrace{4h_1(2^{l-1}+1)\sqrt{\frac{2\sigma^2}{2^{\alpha(l-1)}}\log\left(\frac{1}{\beta}\right)}}_{\text{②}}$$

$$+ \underbrace{4h_1(2^{l-1}+1)\sqrt{\frac{8ds_{l-1}\sigma_0^2}{2^{2\alpha(l-1)}}\log\left(\frac{1}{\beta}\right)}}_{\text{③}}$$

We derive an upper bound for each of the two terms in the above equation.

For ①, we have

$$① \leq 4h_1(2^{l-1}+1)\sqrt{\frac{8d}{2^{(1+\alpha)(l-1)}h_1}\log\left(\frac{1}{\beta}\right)}$$

$$= 8\sqrt{2dh_1\log\left(\frac{1}{\beta}\right)}\left(\sqrt{2^{(1-\alpha)(l-1)}} + \frac{1}{\sqrt{2^{(\alpha+1)(l-1)}}}\right)$$

$$\leq 8\sqrt{2dh_1\log\left(\frac{1}{\beta}\right)}\left(\sqrt{2^{l-1}} + \frac{1}{\sqrt{2^{l-1}}}\right).$$

As to the second term ②, we have

$$② \leq 4h_1(2^{l-1}+1)\sqrt{\frac{2\sigma^2}{2^{\alpha(l-1)}}\log\left(\frac{1}{\beta}\right)}$$

$$= 4h_1\sigma\sqrt{2\log\left(\frac{1}{\beta}\right)}\left(\sqrt{2^{(2-\alpha)(l-1)}} + \frac{1}{\sqrt{2^{\alpha(l-1)}}}\right).$$

Regarding the third term ③, we have

$$③ \leq 4h_1(2^{l-1}+1)\sqrt{\frac{8ds_{l-1}\sigma_0^2}{2^{2\alpha(l-1)}}\log\left(\frac{1}{\beta}\right)}$$

$$\leq 8h_1\sigma_0\sqrt{2dh_1\log\left(\frac{1}{\beta}\right)}\left(2^{(1-\alpha)(l-1)} + \frac{1}{2^{\alpha(l-1)}}\right),$$

where the second inequality is due to $s_l \leq h_1$ for any $l$. Then, the regret in the $l$-th phase $r_l$ is upper bounded by

$$r_l \leq ① + ② + ③.$$

By proper arrangement, we get the result in Eq. (47). $\qquad\square$

**2) Total Regret**

The expected total regret under the DP-DPE algorithm with the central model PRIVATIZER is

$$
\begin{aligned}
\mathbb{E}[R(T)] = {} & O(\sigma T^{1-\alpha/2}\sqrt{d\log(kT)}) \\
& + O(\sigma_0 d^{1+\alpha}T^{1-\alpha}\sqrt{\log(kT)}) \\
& + O(d^{3/2}\sigma_0\sqrt{\log(kT)})
\end{aligned}
\tag{49}
$$

*Proof.* Define $\mathcal{E}_g$ as the event where the "good" event $\mathcal{E}_l$ occurs in every phase, i.e., $\mathcal{E}_g \triangleq \bigcap_{l=1}^{L}\mathcal{E}_l$. It is not difficult to obtain $P\{\mathcal{E}_g\} \geq 1 - 3k\beta L$ by applying union bound. At the same time, let $R_g$ be the regret under event $\mathcal{E}_g$, and $R_b$ be the regret if event $\mathcal{E}_g$ does not hold. Then, the expected total regret in $T$ is $\mathbb{E}[R(T)] = P(\mathcal{E}_g)R_g + (1 - P(\mathcal{E}_g))R_b$.

Under event $\mathcal{E}_g$, the regret in the $l$-th phase $r_l$ satisfies Eq. (47) for any $l \geq 2$. Combining $r_1 \leq 2T_1 \leq 4h_1$ ( since $\langle\theta^*, x^* - x\rangle \leq 2$ for all $x \in \mathcal{D}$), we have

$$
\begin{aligned}
R_g = \sum_{l=1}^{L} r_l \leq {} & 4h_1 + \sum_{l=2}^{L} 8\sqrt{2dh_1\log(1/\beta)} + \left(\sqrt{2^{l-1}} + \frac{1}{\sqrt{2^{l-1}}}\right) \\
& + \sum_{l=2}^{L} 4h_1\sigma\sqrt{2\log(1/\beta)}\left(\sqrt{2^{(2-\alpha)(l-1)}} + \frac{1}{\sqrt{2^{\alpha(l-1)}}}\right) \\
& + \sum_{l=2}^{L} 8h_1\sigma_0\sqrt{2dh_1\log(1/\beta)}\left(2^{(1-\alpha)(l-1)} + \frac{1}{2^{\alpha(l-1)}}\right) \\
\leq {} & 4h_1 + 8\sqrt{2dh_1\log(1/\beta)} \cdot 4\sqrt{2^{L-1}} \\
& + 4h_1\sigma\sqrt{2\log(1/\beta)}\left(\frac{\sqrt{2^{2-\alpha}}}{\sqrt{2^{2-\alpha}}-1} \cdot \sqrt{2^{(L-1)(2-\alpha)}} + C_1\right) \\
& + 8h_1\sigma_0\sqrt{2dh_1\log(1/\beta)}\left(\frac{1}{2^{1-\alpha}-1} \cdot 2^{(L-1)(1-\alpha)} + C_2\right) \\
= {} & 4h_1 + 8\sqrt{2dh_1\log(1/\beta)} \cdot 4\sqrt{2^{L-1}} \\
& + 4h_1\sigma\sqrt{2\log(1/\beta)}\left(4\sqrt{2^{(L-1)(2-\alpha)}} + C_1\right) \\
& + 8h_1\sigma_0\sqrt{2dh_1\log(1/\beta)}\left(\frac{2^{(L-1)(1-\alpha)}}{2^{1-\alpha}-1} + C_1\right)
\end{aligned}
\tag{50}
$$

where $C_1 = \sum_{l=2}^{\infty}\frac{1}{\sqrt{2^{\alpha(l-1)}}}$ and $C_2 = \sum_{l=2}^{\infty}\frac{1}{2^{\alpha(l-1)}} \leq C_1$. Note that $h_L \leq T_L \leq T$, which indicates $2^{L-1} \leq T/h_1$, and $L \leq \log(2T/h_1)$. Then, the above inequality becomes

$$
\begin{aligned}
R_g = \sum_{l=1}^{L} r_l \\
\leq {} & 4h_1 + 8\sqrt{2dh_1\log(1/\beta)} \cdot 4\sqrt{T/h_1} \\
& + 4h_1\sigma\sqrt{2\log(1/\beta)}\left(4\sqrt{(T/h_1)^{2-\alpha}} + C_1\right) \\
& + 8h_1\sigma_0\sqrt{2dh_1\log(1/\beta)}\left(\frac{(T/h_1)^{1-\alpha}}{2^{1-\alpha}-1} + C_1\right) \\
\leq {} & 4h_1 + 32\sqrt{2dT\log(1/\beta)} + 16\sigma\sqrt{2h_1\log(1/\beta)} \cdot T^{1-\alpha/2} \\
& + \frac{8}{2^{1-\alpha}-1}h_1^{1/2+\alpha}\sigma_0\sqrt{2d\log(1/\beta)} \cdot T^{1-\alpha} + 4C_1h_1(\sigma + 2\sigma_0\sqrt{dh_1})\sqrt{2\log(1/\beta)}.
\end{aligned}
\tag{51}
$$

On the other hand, $R_b \leq 2T$ since $\langle \theta^*, x^* - x \rangle \leq 2$ for all $x \in \mathcal{D}$. Choose $\beta = \frac{1}{kT}$ in Algorithm 1. Finally, we have the following results:

$$
\begin{aligned}
\mathbb{E}[R(T)] &= P(\mathcal{E}_g)R_g + (1 - P(\mathcal{E}_g))R_b \\
&\leq R_g + 3k\beta L \cdot 2T \\
&\leq 4h_1 + 32\sqrt{2dT\log(kT)} + 16\sigma\sqrt{2h_1\log(kT)} \cdot T^{1-\alpha/2} \\
&+ 8/(2^{1-\alpha} - 1)h_1^{1/2+\alpha}\sigma_0\sqrt{2d\log(kT)} \cdot T^{1-\alpha} + 4C_1h_1(\sigma + 2\sigma_0\sqrt{dh_1})\sqrt{2\log(kT)} \\
&= O(\sqrt{dT\log(kT)} + O(\sigma T^{1-\alpha/2}\sqrt{d\log(kT)}) + O(\sigma_0 d^{1+\alpha}T^{1-\alpha}\sqrt{\log(kT)}) + O(d^{3/2}\sigma_0\sqrt{\log(kT)}).
\end{aligned}
\tag{52}
$$

$\square$

Finally, substituting $\sigma_0$, we have the total expected regret under the DP-DPE with the central model PRIVATIZER is

$$
\begin{aligned}
\mathbb{E}[R(T)] &= O(\sigma T^{1-\alpha/2}\sqrt{d\log(kT)}) \\
&+ O\left(\frac{Bd^{1+\alpha}\sqrt{\ln(1/\delta)}T^{1-\alpha}\sqrt{\log(kT)}}{\epsilon}\right) \\
&+ O\left(\frac{Bd^{3/2}\sqrt{\ln(1/\delta)}\sqrt{\log(kT)}}{\epsilon}\right)
\end{aligned}
\tag{53}
$$

**3) Communication cost.** Notice that the communicating data in each phase is the local average performance $y_l^u(x)$ for each chosen action $x$ in the support set $\text{supp}(\pi_l)$. Therefore, the total communication cost is

$$
C(T) = \sum_{l=1}^{L} s_l|U_l| \leq \sum_{l=1}^{L}(4d\log\log d + 16) \cdot 2^{\alpha l} = O(dT^\alpha).
\tag{54}
$$

**Theorem 6**(LDP-DPE)  *Consider the Gaussian mechanism with $\sigma_{nl} = \frac{2B\sqrt{2s_l\ln(1.25/\delta)}}{\epsilon}$ in the local DP model. With $\sigma_n = 2\sigma_{nl}\sqrt{\frac{d}{|U_l|}}$ in the $l$-th phase and $\beta = 1/(kT)$, LDP-DPE achieves the following expected regret:*

$$
\begin{aligned}
\mathbb{E}[R(T)] &= O\left(\left(\frac{Bd\sqrt{\ln(1/\delta)}}{\epsilon} + \sigma\right)T^{1-\alpha/2}\sqrt{d\log(kT)}\right) \\
&+ O\left(\frac{Bd^2\sqrt{\ln(1/\delta)}\sqrt{\log(kT)}}{\epsilon}\right)
\end{aligned}
\tag{55}
$$

*and the communication cost is $O(dT^\alpha)$.*

Following a similar line to the proof for Theorem 5, we start with analyzing regret in a specific phase and then combine all phases together to get the total regret.

**1) Regret in a specific phase**

Let $r_l$ denote the incurred regret in the $l$-th phase, i.e., $r_l \triangleq \sum_{t \in \mathcal{T}_l}\langle \theta^*, x^* - x_t \rangle$.

i) First, the total number of pulls in the $l$-th phase is $T_l$, where $T_l = \sum_{x \in \text{supp}(\pi_l)} T_l(x)$. We have

$$
h_l \leq T_l \leq h_l + |\text{supp}(\pi_l)| \leq h_l + h_1,
$$

where $|\text{supp}(\pi_l)| \leq 4d\log\log d + 16 = h_1$.

ii) With $\sigma_n = \frac{2\sigma_{nl}\sqrt{d}}{|U_l|}$, we have $W_l = \sqrt{\frac{4d}{|U_l|h_l}\log\left(\frac{1}{\beta}\right)} + \sqrt{\frac{2\sigma^2}{|U_l|}\log\left(\frac{1}{\beta}\right)} + \sqrt{\frac{8d\sigma_{nl}^2}{|U_l|}\log\left(\frac{1}{\beta}\right)}$. We have the following concentration inequalities, for any $x \in \mathcal{D}$,

$$
P\left\{\langle \tilde{\theta}_l - \theta^*, x \rangle \geq W_l\right\} \leq 3\beta,
\tag{56}
$$

and

$$P\left\{\langle\theta^* - \tilde{\theta}_l, x\rangle \geq W_l\right\} \leq 3\beta. \tag{57}$$

*Proof.* We prove the first concentration inequality in Eq. (56) in the following, and the second inequality can be proved symmetrically. Note that $W_l = W_{l,1} + W_{l,2} + W_{l,3}$, where

$$W_{l,1} = \sqrt{\frac{4d}{h_l|U_l|}\log\left(\frac{1}{\beta}\right)}+, \quad W_{l,2} = \sqrt{\frac{2\sigma^2}{|U_l|}\log\left(\frac{1}{\beta}\right)}, \quad \text{and} \quad W_{l,3} \triangleq \sqrt{\frac{8d\sigma_{nl}^2}{|U_l|}\log\left(\frac{1}{\beta}\right)}.$$

Under the DP-DPE algorithm with the local model PRIVATIZER, the output of $\mathcal{P}$ is, $\tilde{y}_l = \frac{1}{|U_l|}\sum_{u\in U_l}(\vec{y}_l^u + (\gamma_{u,1},\ldots,\gamma_{u,s_l}))$, where $\gamma_{u,j} \overset{i.i.d.}{\sim} \mathcal{N}(0,\sigma_{nl}^2)$. Let $j_x$ denote $j_x$ the index corresponds to the action $x$ in the support set $\text{supp}(\pi_l)$, i.e., $\tilde{y}_l(x) = \frac{1}{|U_l|}\sum_{u\in U_l}(y_l^u(x) + \gamma_{u,j_x})$. Then, the estimated model parameter satisfies

$$\begin{aligned}
\tilde{\theta}_l &= V_l^{-1}G_l \\
&= V_l^{-1}\sum_{x\in\text{supp}(\pi_l)} xT_l(x)\tilde{y}_l(x) \\
&= V_l^{-1}\sum_{x\in\text{supp}(\pi_l)} xT_l(x)\left(\frac{1}{|U_l|}\sum_{u\in U_l}(y_l^u(x) + \gamma_{u,j_x})\right) \\
&= \frac{1}{|U_l|}\sum_{u\in U_l}V_l^{-1}\sum_{x\in\text{supp}(\pi_l)} xT_l(x)y_l^u(x) + \frac{1}{|U_l|}\sum_{u\in U_l}V_l^{-1}\sum_{x\in\text{supp}(\pi_l)} xT_l(x)\gamma_{u,j_x} \\
&= \frac{1}{|U_l|}\sum_{u\in U_l}\theta_u + \frac{1}{|U_l|}\sum_{u\in U_l}V^{-1}\sum_{t\in\mathcal{T}_l}\eta_{u,t}x_t + \frac{1}{|U_l|}\sum_{u\in U_l}V_l^{-1}\sum_{x\in\text{supp}(\pi_l)} xT_l(x)\gamma_{u,j_x}
\end{aligned} \tag{58}$$

For any action $x' \in \mathcal{D}$, the gap between the estimated reward with parameter $\tilde{\theta}_l$ and the true reward with $\theta^*$ satisfies

$$\begin{aligned}
&\langle\tilde{\theta}_l - \theta^*, x'\rangle \\
&= \left\langle\tilde{\theta}_l - \frac{1}{|U_l|}\sum_{u\in U_l}\theta_u + \frac{1}{|U_l|}\sum_{u\in U_l}\theta_u - \theta^*, x'\right\rangle \\
&= \left\langle\tilde{\theta}_l - \frac{1}{|U_l|}\sum_{u\in U_l}\theta_u, x'\right\rangle + \frac{1}{|U_l|}\sum_{u\in U_l}\langle\theta_u - \theta^*, x'\rangle \\
&= \frac{1}{|U_l|}\sum_{u\in U_l}\left\langle V_l^{-1}\sum_{t\in\mathcal{T}_l}\eta_{u,t}x_t, x'\right\rangle + \frac{1}{|U_l|}\sum_{u\in U_l}\left\langle V_l^{-1}\sum_{x\in\text{supp}(\pi_l)} xT_l(x)\gamma_{u,j_x}, x'\right\rangle + \frac{1}{|U_l|}\sum_{u\in U_l}\langle\theta_u - \theta^*, x'\rangle,
\end{aligned}$$

Then, we have

$$\begin{aligned}
&P\left\{\langle\tilde{\theta}_l - \theta^*, x'\rangle \geq W_l\right\} \\
&= P\left\{\frac{1}{|U_l|}\sum_{u\in U_l}\left\langle V_l^{-1}\sum_{t\in\mathcal{T}_l}\eta_{u,t}x_t, x'\right\rangle + \frac{1}{|U_l|}\sum_{u\in U_l}\left\langle V_l^{-1}\sum_{x\in\text{supp}(\pi_l)} xT_l(x)\gamma_{u,j_x}, x'\right\rangle + \frac{1}{|U_l|}\sum_{u\in U_l}\langle\theta_u - \theta^*, x'\rangle \geq W_l\right\} \\
&\leq P\left\{\frac{1}{|U_l|}\sum_{u\in U_l}\left\langle x', V_l^{-1}\sum_{t\in\mathcal{T}_l}\eta_{u,t}x_t\right\rangle \geq W_{l,1}\right\} + P\left\{\frac{1}{|U_l|}\sum_{u\in U_l}\langle\theta_u - \theta^*, x'\rangle \geq W_{l,2}\right\} \\
&\quad + P\left\{\frac{1}{|U_l|}\sum_{u\in U_l}\left\langle x', V_l^{-1}\sum_{x\in\text{supp}(\pi_l)} xT_l(x)\gamma_{u,j_x}\right\rangle \geq W_{l,3}\right\}.
\end{aligned} \tag{59}$$

We have shown that the first and the second probability in the above equation is shown to be less than $\beta$ in Eq. (29) and Eq. (30), respectively. In the following, we try to show that the third term is less than $\beta$.

Note that,

$$\left\langle x', V_l^{-1} \sum_{x \in \mathrm{supp}(\pi_l)} x T_l(x) \gamma_{u,j_x} \right\rangle = \sum_{x \in \mathrm{supp}(\pi_l)} \left\langle x', V_l^{-1} x \right\rangle T_l(x) \gamma_{u,j_x}, \tag{60}$$

and that $\gamma_{u,j} \overset{i.i.d.}{\sim} \mathcal{N}(0, \sigma_{nl}^2)$. The variance (denoted by $\sigma_{u,\mathrm{sum}}^2$) of the above sum of $i.i.d.$ Gaussian variables is

$$\sigma_{u,\mathrm{sum}}^2 = \sum_{x \in \mathrm{supp}(\pi_l)} \left\langle x', V_l^{-1} x \right\rangle^2 T_l(x)^2 \sigma_{nl}^2 \overset{(a)}{\leq} T_l \cdot x'^\top V_l^{-1} \left( \sum_{x \in \mathrm{supp}(\pi_l)} T_l(x) x x^\top \right) V_l^{-1} x' \sigma_{nl}^2 = T_l \|x'\|_{V_l^{-1}}^2 \sigma_{nl}^2,$$

where $(a)$ is from $T_l(x) \leq T_l$ for any $x$ in the support set $\mathrm{supp}(\pi_l)$. Therefore, the LHS of Eq. (60) is a Gaussian variable with variance

$$\sigma_{u,\mathrm{sum}}^2 \leq T_l \|x'\|_{V_l^{-1}}^2 \sigma_{nl}^2 \leq T_l \cdot \frac{2d}{h_l} \cdot \sigma_{nl}^2 \overset{(a)}{\leq} 4 d \sigma_{nl}^2,$$

where $(a)$ is due to $T_l \leq h_l + h_1 \leq 2h_l$. Combining the tail bound for average of $i.i.d.$ Gaussian variables, we have

$$P \left\{ \frac{1}{|U_l|} \sum_{u \in U_l} \left\langle x', V_l^{-1} \sum_{x \in \mathrm{supp}(\pi_l)} x T_l(x) \gamma_{j_x} \right\rangle \geq W_{l,3} \right\} \leq \exp \left\{ -\frac{|U_l| W_{l,3}^2}{2 \sigma_{u,\mathrm{sum}}^2} \right\} \leq \exp \left\{ -\frac{8 d \sigma_{nl}^2 \log(1/\beta)}{8 d \sigma_{nl}^2} \right\} = \beta$$

By now, we complete the proof for Eq. (56). With the symmetrical argument, we obtain the result in Eq. (57). □

iii) Define event $\mathcal{E}_l$ in the $l$-th phase as follows:

$$\mathcal{E}_l \triangleq \left\{ \langle \theta^* - \tilde{\theta}_l, x^* \rangle \leq W_l \quad \text{and} \quad \forall x \in \mathcal{D} \backslash \{x^*\} \quad \langle \tilde{\theta}_l - \theta^*, x \rangle \leq W_l \right\}.$$

It is not difficult to derive $P(\mathcal{E}_l) \geq 1 - 3k\beta$ via union bound.

Then, our second step is to show that under event $\mathcal{E}_l$ for any $l \geq 2$, the regret $r_l \triangleq \sum_{t \in \mathcal{T}_l} \langle \theta^*, x^* - x_t \rangle$ incurred in $l$-th phase satisfies

$$r_l \leq 8\sqrt{2dh_1 \log(1/\beta)} \left( \sqrt{2^{l-1}} + \frac{1}{\sqrt{2^{l-1}}} \right)$$
$$+ 4h_1 \sigma \sqrt{2 \log(1/\beta)} \left( \sqrt{2^{(2-\alpha)(l-1)}} + \frac{1}{\sqrt{2^{\alpha(l-1)}}} \right) \tag{61}$$
$$+ 8h_1 \sigma_0 \sqrt{2dh_1 \log \left( \frac{1}{\beta} \right)} \left( \sqrt{2^{(2-\alpha)(l-1)}} + \frac{1}{\sqrt{2^{\alpha(l-1)}}} \right),$$

where $\sigma_0 = \frac{2B\sqrt{2 \ln(1.25/\delta)}}{\epsilon}$.

*Proof.* To get the bound in Eq. (47), we start with two observations under event $\mathcal{E}_l$:

1. If the optimal action $x^* \in \mathcal{D}_l$, then $x^* \in \mathcal{D}_{l+1}$.

2. For any $x \in \mathcal{D}_{l+1}$, we have $\langle \theta^*, x^* - x \rangle \leq 4W_l$.

**Bound regret $r_l$:**

Note that $h_l = 2^{l-1}h_1$ and $2^{\alpha l} \leq |U_l| \leq 2^{\alpha l} + 1$. We have

1) $W_{l,1} = \sqrt{\frac{4d}{h_l |U_l|} \log\left(\frac{1}{\beta}\right)} \leq \sqrt{\frac{4d}{2^{(1+\alpha)l-1}h_1} \log\left(\frac{1}{\beta}\right)}$;

2) $W_{l,2} = \sqrt{\frac{2\sigma^2}{|U_l|} \log\left(\frac{1}{\beta}\right)} \leq \sqrt{\frac{2\sigma^2}{2^{\alpha l}} \log\left(\frac{1}{\beta}\right)}$;

3) $W_{l,3} = \sqrt{\frac{8d\sigma_{nl}^2}{|U_l|} \log\left(\frac{1}{\beta}\right)} = \sqrt{\frac{8ds_l\sigma_0^2}{|U_l|} \log(\frac{1}{\beta})} \leq \sqrt{\frac{8ds_l\sigma_0^2}{2^{\alpha l}} \log\left(\frac{1}{\beta}\right)}$, where $\sigma_0 = \frac{2B\sqrt{2\ln(1.25/\delta)}}{\epsilon}$.

Combining the upper bound of $T_l$ (i.e., $T_l \leq h_l + h_1$), we derive the regret in any phase $l \geq 2$,

$$
\begin{aligned}
r_l &= \sum_{t=t_l}^{t_l+T_l-1} \langle \theta^*, x^* - x_t \rangle \\
&\leq \sum_{t=t_l}^{t_l+T_l-1} 4W_{l-1} \\
&= \sum_{t=t_l}^{t_l+T_l-1} 4(W_{l-1,1} + W_{l-1,2} + W_{l-1,3}) \\
&= 4T_l(W_{l-1,1} + W_{l-1,2} + W_{l-1,3}) \\
&\leq \underbrace{4h_1(2^{l-1}+1)\sqrt{\frac{4d}{2^{(1+\alpha)(l-1)-1}h_1} \log\left(\frac{1}{\beta}\right)}}_{①} \\
&\quad + \underbrace{4h_1(2^{l-1}+1)\sqrt{\frac{2\sigma^2}{2^{\alpha(l-1)}} \log\left(\frac{1}{\beta}\right)}}_{②} \\
&\quad + \underbrace{4h_1(2^{l-1}+1)\sqrt{\frac{8ds_{l-1}\sigma_0^2}{2^{\alpha(l-1)}} \log\left(\frac{1}{\beta}\right)}}_{③}
\end{aligned}
\tag{62}
$$

We derive an upper bound for each of the two terms in the above equation.

For ①, we have

$$
\begin{aligned}
① &\leq 4h_1(2^{l-1}+1)\sqrt{\frac{8d}{2^{(1+\alpha)(l-1)}h_1} \log\left(\frac{1}{\beta}\right)} \\
&= 8\sqrt{2dh_1 \log\left(\frac{1}{\beta}\right)} \left(\sqrt{2^{(1-\alpha)(l-1)}} + \frac{1}{\sqrt{2^{(\alpha+1)(l-1)}}}\right) \\
&\leq 8\sqrt{2dh_1 \log\left(\frac{1}{\beta}\right)} \left(\sqrt{2^{l-1}} + \frac{1}{\sqrt{2^{l-1}}}\right).
\end{aligned}
$$

As to the second term ②, we have

$$
\begin{aligned}
② &\leq 4h_1(2^{l-1}+1)\sqrt{\frac{2\sigma^2}{2^{\alpha(l-1)}} \log\left(\frac{1}{\beta}\right)} \\
&= 4h_1\sigma\sqrt{2\log\left(\frac{1}{\beta}\right)} \left(\sqrt{2^{(2-\alpha)(l-1)}} + \frac{1}{\sqrt{2^{\alpha(l-1)}}}\right).
\end{aligned}
$$

Regarding the third term ③, we have

$$
\text{③} \leq 4h_1(2^{l-1}+1)\sqrt{\frac{8ds_{l-1}\sigma_0^2}{2^{\alpha(l-1)}}\log\left(\frac{1}{\beta}\right)}
$$

$$
\leq 8h_1\sigma_0\sqrt{2dh_1\log\left(\frac{1}{\beta}\right)}\left(\sqrt{2^{(2-\alpha)(l-1)}}+\frac{1}{\sqrt{2^{\alpha(l-1)}}}\right),
$$

where the second inequality is due to $s_l \leq h_1$ for any $l$. Then, the regret in the $l$-th phase $r_l$ is upper bounded by

$$
r_l \leq \text{①} + \text{②} + \text{③}.
$$

By proper arrangement, we get the result in Eq. (61). □

## 2) Total Regret

The expected total regret under the DP-DPE algorithm with the central model PRIVATIZER is

$$
\mathbb{E}[R(T)] = O(\sigma T^{1-\alpha/2}\sqrt{d\log(kT)})
$$
$$
+ O(\sigma_0 d^{3/2} T^{1-\alpha/2}\sqrt{\log(kT)}) \tag{63}
$$
$$
+ O(d^{3/2}\sigma_0\sqrt{\log(kT)})
$$

*Proof.* Define $\mathcal{E}_g$ as the event where the "good" event $\mathcal{E}_l$ occurs in every phase, i.e., $\mathcal{E}_g \triangleq \bigcap_{l=1}^L \mathcal{E}_l$. It is not difficult to obtain $P\{\mathcal{E}_g\} \geq 1 - 3k\beta L$ by applying union bound. At the same time, let $R_g$ denote the regret under event $\mathcal{E}_g$, and $R_b$ be the regret if event $\mathcal{E}_g$ does not hold. Then, the expected total regret in $T$ is $\mathbb{E}[R(T)] = P(\mathcal{E}_g)R_g + (1 - P(\mathcal{E}_g))R_b$.

Under event $\mathcal{E}_g$, the regret in the $l$-th phase $r_l$ satisfies Eq. (61) for any $l \geq 2$. Combining $r_1 \leq 2T_1 \leq 4h_1$ ( since $\langle \theta^*, x^* - x \rangle \leq 2$ for all $x \in \mathcal{D}$), we have

$$
R_g = \sum_{l=1}^L r_l
$$
$$
\leq 4h_1 + \sum_{l=2}^L 8\sqrt{2dh_1\log(1/\beta)+}\left(\sqrt{2^{l-1}}+\frac{1}{\sqrt{2^{l-1}}}\right)
$$
$$
+ \sum_{l=2}^L 4h_1\sigma\sqrt{2\log(1/\beta)}\left(\sqrt{2^{(2-\alpha)(l-1)}}+\frac{1}{\sqrt{2^{\alpha(l-1)}}}\right)
$$
$$
+ \sum_{l=2}^L 8h_1\sigma_0\sqrt{2dh_1\log\left(\frac{1}{\beta}\right)}\left(\sqrt{2^{(2-\alpha)(l-1)}}+\frac{1}{\sqrt{2^{\alpha(l-1)}}}\right) \tag{64}
$$
$$
\leq 4h_1 + 8\sqrt{2dh_1\log(1/\beta)}\cdot 4\sqrt{2^{L-1}}
$$
$$
+ 4h_1(\sigma + 2\sigma_0\sqrt{dh_1})\sqrt{2\log(1/\beta)}\left(\frac{\sqrt{2^{2-\alpha}}}{\sqrt{2^{2-\alpha}}-1}\cdot\sqrt{2^{(L-1)(2-\alpha)}}+C_1\right)
$$
$$
\leq 4h_1 + 8\sqrt{2dh_1\log(1/\beta)}\cdot 4\sqrt{2^{L-1}}
$$
$$
+ 4h_1(\sigma + 2\sigma_0\sqrt{dh_1})\sqrt{2\log(1/\beta)}\left(4\sqrt{2^{(L-1)(2-\alpha)}}+C_1\right)
$$

where $C_1 = \sum_{l=2}^{\infty} \frac{1}{\sqrt{2^{\alpha(l-1)}}}$. Note that $h_L \leq T_L \leq T$, which indicates $2^{L-1} \leq T/h_1$, and $L \leq \log(2T/h_1)$. Then, the above inequality becomes

$$
\begin{aligned}
R_g &= \sum_{l=1}^{L} r_l \\
&\leq 4h_1 + 8\sqrt{2dh_1 \log(1/\beta)} \cdot 4\sqrt{T/h_1} \\
&\quad + 4h_1(\sigma + 2\sigma_0\sqrt{dh_1})\sqrt{2\log(1/\beta)}\left(4\sqrt{(T/h_1)^{2-\alpha}} + C_1\right) \\
&\leq 4h_1 + 32\sqrt{2dT\log(1/\beta)} + 16(\sigma + 2\sigma_0\sqrt{dh_1})\sqrt{2h_1 \log(1/\beta)} \cdot T^{1-\alpha/2} \\
&\quad + 4C_1 h_1(\sigma + 2\sigma_0\sqrt{dh_1})\sqrt{2\log(1/\beta)}.
\end{aligned}
\tag{65}
$$

On the other hand, $R_b \leq 2T$ since $\langle \theta^*, x^* - x \rangle \leq 2$ for all $x \in \mathcal{D}$. Choose $\beta = \frac{1}{kT}$ in Algorithm 1. Finally, we have the following results:

$$
\begin{aligned}
\mathbb{E}[R(T)] &= P(\mathcal{E}_g)R_g + (1 - P(\mathcal{E}_g))R_b \\
&\leq R_g + 3k\beta L \cdot 2T \\
&\leq 4h_1 + 32\sqrt{2dT\log(kT)} + 16\sigma\sqrt{2h_1\log(kT)} \cdot T^{1-\alpha/2} \\
&\quad + 32\sigma_0 h_1 \sqrt{2d\log(kT)} \cdot T^{1-\alpha/2} + 4C_1 h_1(\sigma + 2\sigma_0\sqrt{dh_1})\sqrt{2\log(kT)} \\
&= O(\sqrt{dT\log(kT)} + O(\sigma T^{1-\alpha/2}\sqrt{d\log(kT)}) + O(\sigma_0 d^{3/2}T^{1-\alpha/2}\sqrt{\log(kT)}) + O(d^2\sigma_0\sqrt{\log(kT)}).
\end{aligned}
\tag{66}
$$

$\square$

Finally, substituting $\sigma_0$, we have the total expected regret under the DP-DPE with the local model PRIVATIZER is

$$
\begin{aligned}
\mathbb{E}[R(T)] &= O\left(\left(\frac{Bd\sqrt{\ln(1/\delta)}}{\epsilon} + \sigma\right)T^{1-\alpha/2}\sqrt{d\log(kT)}\right) \\
&\quad + O\left(\frac{Bd^2\sqrt{\ln(1/\delta)}\sqrt{\log(kT)}}{\epsilon}\right)
\end{aligned}
\tag{67}
$$

**3) Communication cost.**

Notice that the communicating data in each phase is the local average performance $y_l^u(x)$ for each chosen action $x$ in the support set $\text{supp}(\pi_l)$. Therefore, the total communication cost is

$$
C(T) = \sum_{l=1}^{L} s_l |U_l| \leq \sum_{l=1}^{L} (4d\log\log d + 16) \cdot 2^{\alpha l} = O(dT^\alpha).
\tag{68}
$$

### B.3 Proof of Theorem 7

**Theorem 7** (SDP-DPE). *With $\sigma_n = 2\sigma_{ns}\sqrt{d} = O\left(\frac{B\sqrt{ds_l}\log(s_l/\delta)}{\epsilon|U_l|}\right)$ in the $l$-th phase and $\beta = 1/(kT)$, the SDP-DPE achieves the following expected regret:*

$$
\begin{aligned}
\mathbb{E}[R(T)] &= O(\sigma T^{1-\alpha/2}\sqrt{d\log(kT)}) \\
&\quad + O\left(\frac{Bd^{1+\alpha}\ln(d/\delta)T^{1-\alpha}\sqrt{\log(kT)}}{\epsilon}\right) \\
&\quad + O\left(\frac{Bd^{3/2}\ln(d/\delta)\sqrt{\log(kT)}}{\epsilon}\right)
\end{aligned}
\tag{69}
$$

*and the communication cost is $O(dT^{3\alpha}/2)$.*

*Proof.* Following a similar line to the proof for Theorem 5, we start with analyzing regret in a specific phase and then combine all phases together to get the total regret.

**1) Regret in a specific phase**

Let $r_l$ denote the incurred regret in the $l$-th phase, i.e., $r_l \triangleq \sum_{t \in \mathcal{T}_l} \langle \theta^*, x^* - x_t \rangle$.

i) First, the total number of pulls in the $l$-th phase $T_l$ satisfies

$$h_l \leq T_l \leq h_l + |\text{supp}(\pi_l)| \leq h_l + h_1,$$

where $|\text{supp}(\pi_l)| \leq 4d \log \log d + 16 = h_1$.

ii) With $\sigma_n = 2\sigma_{ns}\sqrt{d}$, we have $W_l = \sqrt{\frac{4d}{|U_l|h_l} \log\left(\frac{1}{\beta}\right)} + \sqrt{\frac{2\sigma^2}{|U_l|} \log\left(\frac{1}{\beta}\right)} + \sqrt{8d\sigma_{ns}^2 \log\left(\frac{1}{\beta}\right)}$. We have the following concentration inequalities, for any $x \in \mathcal{D}$,

$$P\left\{ \langle \tilde{\theta}_l - \theta^*, x \rangle \geq W_l \right\} \leq 3\beta, \tag{70}$$

and

$$P\left\{ \langle \theta^* - \tilde{\theta}_l, x \rangle \geq W_l \right\} \leq 3\beta. \tag{71}$$

*Proof.* We prove the first concentration inequality in Eq. (70) in the following, and the second inequality can be proved symmetrically. Note that $W_l = W_{l,1} + W_{l,2} + W_{l,3}$, where

$$W_{l,1} = \sqrt{\frac{4d}{h_l|U_l|} \log\left(\frac{1}{\beta}\right)}+, \quad W_{l,2} = \sqrt{\frac{2\sigma^2}{|U_l|} \log\left(\frac{1}{\beta}\right)}, \quad \text{and} \quad W_{l,3} \triangleq \sqrt{8d\sigma_{ns}^2 \log\left(\frac{1}{\beta}\right)}.$$

Under the DP-DPE algorithm with the shuffle model PRIVATIZER, the output of $\mathcal{P}$ is, $\tilde{y}_l = (\mathcal{A} \circ \mathcal{S} \circ \mathcal{R}^{|U_l|})(\{\vec{y}_l^u\}_{u \in U_l}) = \mathcal{A}(\mathcal{S}(\{\mathcal{R}(\vec{y}_l^u)\}_{u \in U_l}))$, where $\mathcal{A}, \mathcal{S}$ and $\mathcal{R}$ follow Algorithm 2. From Theorem 9, we know that the output of $(\mathcal{A} \circ \mathcal{S} \circ \mathcal{R}^{|U_l|})(\{\vec{y}_l^u\}_{u \in U_l})$ is an unbiased estimator of the average of the $|U_l|$ input vectors $\{\vec{y}_l^u\}_{u \in U_l}$ and that the error distribution is sub-Gaussian with variance $\sigma_{ns}^2 = O\left(\frac{B^2 s_l \ln^2(s_l/\delta)}{\epsilon^2 |U_l|^2}\right)$. Let $\gamma^{(e)}$ denote the error vector and $j_x$ represent the index corresponding to the action $x$ in the support set $\text{supp}(\pi_l)$. That is, $\gamma_{j_x}^{(e)} = \tilde{y}_l(x) - \frac{1}{|U_l|} \sum_{u \in U_l} y_l^u(x)$ is $\sigma_{ns}$-sub-Gaussian with $\mathbb{E}[\gamma_{j_x}^{(e)}] = 0$. The estimated model parameter satisfies

$$
\begin{aligned}
\tilde{\theta}_l &= V_l^{-1} G_l \\
&= V_l^{-1} \sum_{x \in \text{supp}(\pi_l)} x T_l(x) \tilde{y}_l(x) \\
&= V_l^{-1} \sum_{x \in \text{supp}(\pi_l)} x T_l(x) \left( \frac{1}{|U_l|} \sum_{u \in U_l} y_l^u(x) + \gamma_{j_x}^{(e)} \right) \\
&= \frac{1}{|U_l|} \sum_{u \in U_l} V_l^{-1} \sum_{x \in \text{supp}(\pi_l)} x T_l(x) y_l^u(x) + V_l^{-1} \sum_{x \in \text{supp}(\pi_l)} x T_l(x) \gamma_{j_x}^{(e)} \\
&= \frac{1}{|U_l|} \sum_{u \in U_l} \theta_u + \frac{1}{|U_l|} \sum_{u \in U_l} V^{-1} \sum_{t \in \mathcal{T}_l} \eta_{u,t} x_t + V_l^{-1} \sum_{x \in \text{supp}(\pi_l)} x T_l(x) \gamma_{j_x}^{(e)}
\end{aligned} \tag{72}
$$

For any action $x' \in \mathcal{D}$, the gap between the estimated reward with parameter $\tilde{\theta}_l$ and the true reward with $\theta^*$ satisfies

$$\langle \tilde{\theta}_l - \theta^*, x' \rangle$$

$$= \left\langle \tilde{\theta}_l - \frac{1}{|U_l|} \sum_{u \in U_l} \theta_u + \frac{1}{|U_l|} \sum_{u \in U_l} \theta_u - \theta^*, x' \right\rangle$$

$$= \left\langle \tilde{\theta}_l - \frac{1}{|U_l|} \sum_{u \in U_l} \theta_u, x' \right\rangle + \frac{1}{|U_l|} \sum_{u \in U_l} \langle \theta_u - \theta^*, x' \rangle$$

$$= \frac{1}{|U_l|} \sum_{u \in U_l} \left\langle V_l^{-1} \sum_{t \in \mathcal{T}_l} \eta_{u,t} x_t, x' \right\rangle + \left\langle V_l^{-1} \sum_{x \in \mathrm{supp}(\pi_l)} x T_l(x) \gamma_{j_x}^{(e)}, x' \right\rangle + \frac{1}{|U_l|} \sum_{u \in U_l} \langle \theta_u - \theta^*, x' \rangle,$$

Then, we have

$$P\left\{ \langle \tilde{\theta}_l - \theta^*, x \rangle \geq W_l \right\}$$

$$= P\left\{ \frac{1}{|U_l|} \sum_{u \in U_l} \left\langle V_l^{-1} \sum_{t \in \mathcal{T}_l} \eta_{u,t} x_t, x' \right\rangle + \left\langle V_l^{-1} \sum_{x \in \mathrm{supp}(\pi_l)} x T_l(x) \gamma_{j_x}^{(e)}, x' \right\rangle + \frac{1}{|U_l|} \sum_{u \in U_l} \langle \theta_u - \theta^*, x' \rangle \geq W_l \right\}$$

$$= P\left\{ \frac{1}{|U_l|} \sum_{u \in U_l} \left\langle V_l^{-1} \sum_{t \in \mathcal{T}_l} \eta_{u,t} x_t, x' \right\rangle + \left\langle V_l^{-1} \sum_{x \in \mathrm{supp}(\pi_l)} x T_l(x) \gamma_{j_x}^{(e)}, x' \right\rangle + \frac{1}{|U_l|} \sum_{u \in U_l} \langle \theta_u - \theta^*, x' \rangle \geq W_{l,1} + W_{l,2} + W_{l,3} \right\}$$

$$\leq P\left\{ \frac{1}{|U_l|} \sum_{u \in U_l} \left\langle x', V_l^{-1} \sum_{t \in \mathcal{T}_l} \eta_{u,t} x_t \right\rangle \geq W_{l,1} \right\} + P\left\{ \frac{1}{|U_l|} \sum_{u \in U_l} \langle \theta_u - \theta^*, x \rangle \geq W_{l,2} \right\}$$

$$+ P\left\{ \left\langle x', V_l^{-1} \sum_{x \in \mathrm{supp}(\pi_l)} x T_l(x) \gamma_{j_x}^{(e)} \right\rangle \geq W_{l,3} \right\}.$$

$$(73)$$

We have shown that the first and the second probability in the above equation is shown to be less than $\beta$ in Eq. (29) and Eq. (30), respectively. In the following, we try to show that the third term is less than $\beta$.

Note that $\gamma_{j_x}^{(e)}$ is $\sigma_{ns}$-sub-Gaussian and *i.i.d.* over each coordinate in $\{j_x : x \in \mathrm{supp}(\pi_l)\}$, and that

$$\left\langle x', V_l^{-1} \sum_{x \in \mathrm{supp}(\pi_l)} x T_l(x) \gamma_{j_x}^{(e)} \right\rangle = \sum_{x \in \mathrm{supp}(\pi_l)} \langle x', V_l^{-1} x \rangle T_l(x) \gamma_{j_x}^{(e)}. \qquad (74)$$

The variance (denoted by $\sigma_{\mathrm{sub\text{-}G}}^2$) of the above sum of *i.i.d.* sub-Gaussian variables is

$$\sigma_{\mathrm{sub\text{-}G}}^2 = \sum_{x \in \mathrm{supp}(\pi_l)} \langle x', V_l^{-1} x \rangle^2 T_l(x)^2 \sigma_{ns}^2 \overset{(a)}{\leq} T_l \cdot x'^\top V_l^{-1} \left( \sum_{x \in \mathrm{supp}(\pi_l)} T_l(x) x x^\top \right) V_l^{-1} x' \sigma_{ns}^2 = T_l \|x'\|_{V_l^{-1}}^2 \sigma_{ns}^2,$$

where $(a)$ is from $T_l(x) \leq T_l$ for any $x$ in the support set $\mathrm{supp}(\pi_l)$. Therefore, the LHS of Eq. (74) is $\sigma_{\mathrm{sub\text{-}G}}$-sub-Gaussian with variance proxy

$$\sigma_{\mathrm{sub\text{-}G}}^2 \leq T_l \|x'\|_{V_l^{-1}}^2 \sigma_{ns}^2 \leq T_l \cdot \frac{2d}{h_l} \cdot \sigma_{ns}^2 \overset{(a)}{\leq} 4d\sigma_{ns}^2,$$

where $(a)$ is due to $T_l \leq h_l + h_1 \leq 2h_l$. Combining the property for sub-Gaussian variables, we have

$$P\left\{ \left\langle x', V_l^{-1} \sum_{x \in \mathrm{supp}(\pi_l)} x T_l(x) \gamma_{j_x}^{(e)} \right\rangle \geq W_{l,3} \right\} \leq \exp\left\{ -\frac{W_{l,3}^2}{2\sigma_{\mathrm{sub\text{-}G}}^2} \right\} \leq \exp\left\{ -\frac{8d\sigma_{ns}^2 \log(1/\beta)}{8d\sigma_{ns}^2} \right\} = \beta$$

By now, we complete the proof for Eq. (42). With the symmetrical argument, we obtain the result in Eq. (43). □

iii) Define event $\mathcal{E}_l$ in the $l$-th phase as follows:

$$\mathcal{E}_l \triangleq \left\{ \langle \theta^* - \tilde{\theta}_l, x^* \rangle \leq W_l \quad \text{and} \quad \forall x \in \mathcal{D} \backslash \{x^*\} \quad \langle \tilde{\theta}_l - \theta^*, x \rangle \leq W_l \right\}.$$

It is not difficult to derive $P(\mathcal{E}_l) \geq 1 - 3k\beta$ via union bound.

Then, our second step is to show that under event $\mathcal{E}_l$ for any $l \geq 2$, the regret $r_l \triangleq \sum_{t \in \mathcal{T}_l} \langle \theta^*, x^* - x_t \rangle$ incurred in $l$-th phase satisfies

$$
\begin{aligned}
r_l \leq & 8\sqrt{2dh_1 \log(1/\beta)} \left( \sqrt{2^{l-1}} + \frac{1}{\sqrt{2^{l-1}}} \right) \\
& + 4h_1 \sigma \sqrt{2\log(1/\beta)} \left( \sqrt{2^{(2-\alpha)(l-1)}} + \frac{1}{\sqrt{2^{\alpha(l-1)}}} \right) \\
& + 8h_1 \sigma_0 \sqrt{2dh_1 \log\left(\frac{1}{\beta}\right)} \left( 2^{(1-\alpha)(l-1)} + \frac{1}{2^{\alpha(l-1)}} \right),
\end{aligned}
\tag{75}
$$

where $\sigma_0 = \frac{CB \ln(h_1/\delta)}{\epsilon}$.

*Proof.* To get the bound in Eq. (75) , we start with two observations under event $\mathcal{E}_l$:

1. If the optimal action $x^* \in \mathcal{D}_l$, then $x^* \in \mathcal{D}_{l+1}$.

2. For any $x \in \mathcal{D}_{l+1}$, we have $\langle \theta^*, x^* - x \rangle \leq 4W_l$.

**Bound regret $r_l$:**

Note that $h_l = 2^{l-1}h_1$, $s_l \leq h_1$, and $2^{\alpha l} \leq |U_l| \leq 2^{\alpha l} + 1$. Hence,

1) $W_{l,1} = \sqrt{\frac{4d}{h_l |U_l|} \log\left(\frac{1}{\beta}\right)} \leq \sqrt{\frac{4d}{2^{(1+\alpha)l-1}h_1} \log\left(\frac{1}{\beta}\right)}$;

2) $W_{l,2} = \sqrt{\frac{2\sigma^2}{|U_l|} \log\left(\frac{1}{\beta}\right)} \leq \sqrt{\frac{2\sigma^2}{2^{\alpha l}} \log\left(\frac{1}{\beta}\right)}$;

3) Besides, we let $\sigma_{ns} = \frac{CB\sqrt{s_l}\ln(s_l/\delta)}{\epsilon |U_l|}$ and have

$$
\begin{aligned}
W_{l,3} &= \sqrt{8d\sigma_{ns}^2 \log\left(\frac{1}{\beta}\right)} \\
&= \sqrt{\frac{8dC^2 B^2 s_l \ln^2(s_l/\delta)}{\epsilon^2 |U_l|^2} \log\left(\frac{1}{\beta}\right)} \\
&\overset{(a)}{\leq} \sqrt{\frac{8dC^2 B^2 h_1 \ln^2(h_1/\delta)}{\epsilon^2 |U_l|^2} \log\left(\frac{1}{\beta}\right)} \\
&\overset{(b)}{\leq} \sqrt{\frac{8dh_1 \sigma_0^2}{2^{2\alpha l}} \log\left(\frac{1}{\beta}\right)}
\end{aligned}
$$

where $(a)$ is due to $s_l \leq h_1$ and $(b)$ is from $\sigma_0 \triangleq \frac{CB \ln(h_1/\delta)}{\epsilon}$ and $|U_l| \geq 2^{\alpha l}$.

Combining the upper bound of $T_l$ (i.e., $T_l \leq h_l + h_1$), we derive the regret in any phase $l \geq 2$,

$$
\begin{aligned}
r_l &= \sum_{t=t_l}^{t_l+T_l-1} \langle \theta^*, x^* - x_t \rangle \\
&\leq \sum_{t=t_l}^{t_l+T_l-1} 4W_{l-1} \\
&= \sum_{t=t_l}^{t_l+T_l-1} 4(W_{l-1,1} + W_{l-1,2} + W_{l-1,3}) \\
&= 4T_l(W_{l-1,1} + W_{l-1,2} + W_{l-1,3}) \\
&\leq \underbrace{4h_1(2^{l-1}+1)\sqrt{\frac{4d}{2^{(1+\alpha)(l-1)-1}h_1}\log\left(\frac{1}{\beta}+\right)}}_{\text{①}} \\
&\quad + \underbrace{4h_1(2^{l-1}+1)\sqrt{\frac{2\sigma^2}{2^{\alpha(l-1)}}\log\left(\frac{1}{\beta}\right)}}_{\text{②}} \\
&\quad + \underbrace{4h_1(2^{l-1}+1)\sqrt{\frac{8dh_1\sigma_0^2}{2^{2\alpha(l-1)}}\log\left(\frac{1}{\beta}\right)}}_{\text{③}}
\end{aligned}
\tag{76}
$$

We derive an upper bound for each of the two terms in the above equation.

For ①, we have

$$
\begin{aligned}
\text{①} &\leq 4h_1(2^{l-1}+1)\sqrt{\frac{8d}{2^{(1+\alpha)(l-1)}h_1}\log\left(\frac{1}{\beta}\right)} \\
&= 8\sqrt{2dh_1\log\left(\frac{1}{\beta}\right)}\left(\sqrt{2^{(1-\alpha)(l-1)}} + \frac{1}{\sqrt{2^{(\alpha+1)(l-1)}}}\right) \\
&\leq 8\sqrt{2dh_1\log\left(\frac{1}{\beta}\right)}\left(\sqrt{2^{l-1}} + \frac{1}{\sqrt{2^{l-1}}}\right).
\end{aligned}
$$

As to the second term ②, we have

$$
\begin{aligned}
\text{②} &\leq 4h_1(2^{l-1}+1)\sqrt{\frac{2\sigma^2}{2^{\alpha(l-1)}}\log\left(\frac{1}{\beta}\right)} \\
&= 4h_1\sigma\sqrt{2\log\left(\frac{1}{\beta}\right)}\left(\sqrt{2^{(2-\alpha)(l-1)}} + \frac{1}{\sqrt{2^{\alpha(l-1)}}}\right).
\end{aligned}
$$

Regarding the third term ③, we have

$$
\begin{aligned}
\text{③} &\leq 4h_1(2^{l-1}+1)\sqrt{\frac{8dh_1\sigma_0^2}{2^{2\alpha(l-1)}}\log\left(\frac{1}{\beta}\right)} \\
&= 8h_1\sigma_0\sqrt{2dh_1\log\left(\frac{1}{\beta}\right)}\left(2^{(1-\alpha)(l-1)} + \frac{1}{2^{\alpha(l-1)}}\right).
\end{aligned}
$$

Then, the regret in the $l$-th phase $r_l$ is upper bounded by

$$
r_l \leq \text{①} + \text{②} + \text{③}.
$$

By proper arrangement, we get the result in Eq. (75). □

### 2) Total Regret

The expected total regret under the DP-DPE algorithm with the shuffle model PRIVATIZER is

$$
\begin{aligned}
\mathbb{E}[R(T)] &= O(\sigma T^{1-\alpha/2}\sqrt{d\log(kT)}) \\
&+ O(\sigma_0 d^{1+\alpha}T^{1-\alpha}\sqrt{\log(kT)}) \\
&+ O(d^{3/2}\sigma_0\sqrt{\log(kT)})
\end{aligned}
\tag{77}
$$

*Proof.* Define $\mathcal{E}_g$ as the event where the "good" event $\mathcal{E}_l$ occurs in every phase, i.e., $\mathcal{E}_g \triangleq \bigcap_{l=1}^{L}\mathcal{E}_l$. It is not difficult to obtain $P\{\mathcal{E}_g\} \geq 1 - 3k\beta L$ by applying union bound. At the same time, let $R_g$ be the regret under event $\mathcal{E}_g$, and $R_b$ be the regret if event $\mathcal{E}_g$ does not hold. Then, the expected total regret in $T$ is $\mathbb{E}[R(T)] = P(\mathcal{E}_g)R_g + (1 - P(\mathcal{E}_g))R_b$.

Under event $\mathcal{E}_g$, the regret in the $l$-th phase $r_l$ satisfies Eq. (75) for any $l \geq 2$. Combining $r_1 \leq 2T_1 \leq 4h_1$ ( since $\langle\theta^*, x^* - x\rangle \leq 2$ for all $x \in \mathcal{D}$), we have

$$
\begin{aligned}
R_g = \sum_{l=1}^{L} r_l &\leq 4h_1 + \sum_{l=2}^{L} 8\sqrt{2dh_1\log(1/\beta)} + \left(\sqrt{2^{l-1}} + \frac{1}{\sqrt{2^{l-1}}}\right) \\
&+ \sum_{l=2}^{L} 4h_1\sigma\sqrt{2\log(1/\beta)}\left(\sqrt{2^{(2-\alpha)(l-1)}} + \frac{1}{\sqrt{2^{\alpha(l-1)}}}\right) \\
&+ \sum_{l=2}^{L} 8h_1\sigma_0\sqrt{2dh_1\log(1/\beta)}\left(2^{(1-\alpha)(l-1)} + \frac{1}{2^{\alpha(l-1)}}\right) \\
&\leq 4h_1 + 8\sqrt{2dh_1\log(1/\beta)}\cdot 4\sqrt{2^{L-1}} \\
&+ 4h_1\sigma\sqrt{2\log(1/\beta)}\left(\frac{\sqrt{2^{2-\alpha}}}{\sqrt{2^{2-\alpha}}-1}\cdot\sqrt{2^{(L-1)(2-\alpha)}} + C_1\right) \\
&+ 8h_1\sigma_0\sqrt{2dh_1\log(1/\beta)}\left(\frac{1}{2^{1-\alpha}-1}\cdot 2^{(L-1)(1-\alpha)} + C_2\right) \\
&= 4h_1 + 8\sqrt{2dh_1\log(1/\beta)}\cdot 4\sqrt{2^{L-1}} \\
&+ 4h_1\sigma\sqrt{2\log(1/\beta)}\left(4\sqrt{2^{(L-1)(2-\alpha)}} + C_1\right) \\
&+ 8h_1\sigma_0\sqrt{2dh_1\log(1/\beta)}\left(\frac{2^{(L-1)(1-\alpha)}}{2^{1-\alpha}-1} + C_1\right)
\end{aligned}
\tag{78}
$$

where $C_1 = \sum_{l=2}^{\infty}\frac{1}{\sqrt{2^{\alpha(l-1)}}}$ and $C_2 = \sum_{l=2}^{\infty}\frac{1}{2^{\alpha(l-1)}} \leq C_1$. Note that $h_L \leq T_L \leq T$, which indicates $2^{L-1} \leq T/h_1$, and $L \leq \log(2T/h_1)$. Then, the above inequality becomes

$$
\begin{aligned}
R_g = \sum_{l=1}^{L} r_l \\
\leq 4h_1 + 8\sqrt{2dh_1\log(1/\beta)}\cdot 4\sqrt{T/h_1} \\
+ 4h_1\sigma\sqrt{2\log(1/\beta)}\left(4\sqrt{(T/h_1)^{2-\alpha}} + C_1\right) \\
+ 8h_1\sigma_0\sqrt{2dh_1\log(1/\beta)}\left(\frac{(T/h_1)^{1-\alpha}}{2^{1-\alpha}-1} + C_1\right) \\
\leq 4h_1 + 32\sqrt{2dT\log(1/\beta)} + 16\sigma\sqrt{2h_1\log(1/\beta)}\cdot T^{1-\alpha/2} \\
+ \frac{8}{2^{1-\alpha}-1}h_1^{1/2+\alpha}\sigma_0\sqrt{2d\log(1/\beta)}\cdot T^{1-\alpha} + 4C_1 h_1(\sigma + 2\sigma_0\sqrt{dh_1})\sqrt{2\log(1/\beta)}.
\end{aligned}
\tag{79}
$$

On the other hand, $R_b \leq 2T$ since $\langle \theta^*, x^* - x \rangle \leq 2$ for all $x \in \mathcal{D}$. Choose $\beta = \frac{1}{kT}$ in Algorithm 1. Finally, we have the following results:

$$
\begin{aligned}
\mathbb{E}[R(T)] &= P(\mathcal{E}_g)R_g + (1 - P(\mathcal{E}_g))R_b \\
&\leq R_g + 3k\beta L \cdot 2T \\
&\leq 4h_1 + 32\sqrt{2dT\log(kT)} + 16\sigma\sqrt{2h_1\log(kT)} \cdot T^{1-\alpha/2} \\
&\quad + 8/(2^{1-\alpha} - 1)h_1^{1/2+\alpha}\sigma_0\sqrt{2d\log(kT)} \cdot T^{1-\alpha} + 4C_1 h_1(\sigma + 2\sigma_0\sqrt{dh_1})\sqrt{2\log(kT)} \\
&= O(\sqrt{dT\log(kT)}) + O(\sigma T^{1-\alpha/2}\sqrt{d\log(kT)}) + O(\sigma_0 d^{1+\alpha}T^{1-\alpha}\sqrt{\log(kT)}) + O(d^{3/2}\sigma_0\sqrt{\log(kT)}).
\end{aligned}
\tag{80}
$$

$\square$

Finally, substituting $\sigma_0$, we have the total expected regret under the DP-DPE with the shuffle model PRIVATIZER is

$$
\begin{aligned}
\mathbb{E}[R(T)] &= O(\sigma T^{1-\alpha/2}\sqrt{d\log(kT)}) \\
&\quad + O\left(\frac{Bd^{1+\alpha}\ln(d/\delta)T^{1-\alpha}\sqrt{\log(kT)}}{\epsilon}\right) \\
&\quad + O\left(\frac{Bd^{3/2}\ln(d/\delta)\sqrt{\log(kT)}}{\epsilon}\right)
\end{aligned}
\tag{81}
$$

**3) Communication cost.** The communicate cost in the shuffle model is slightly different from the central model and the local model because it communicates $(g + b)s_l$ bits from each participating client in the $l$-th phase instead of $d$-dimensional vectors. Based on our setting (Eq. (22) in Algorithm 2), we have $(g + b) = \max\left\{O\left(\frac{\sqrt{|U_l|}}{\ln(d)}\right), O\left(\sqrt{d} + \frac{d\ln(d)}{|U_l|}\right), O\left(\frac{\ln(d)}{|U_l|}\right)\right\}$. Combining $s_l \leq h_1$, the total communication cost is

$$
\sum_{l=1}^{L}(g + b)s_l|U_l| = O\left(\sum_{l=1}^{L}\frac{2^{\frac{3}{2}\alpha l}h_1}{\ln(d)}\right) = O\left(dT^{(3/2)\alpha}\right).
$$

$\square$

