# OpenReview forum: "Global Reward Maximization with Partial Feedback via Differentially Private Distributed Linear Bandits"
_TMLR — Withdrawn by Authors_

### Review · Reviewer_vQzp · 2022-04-18

**Summary Of Contributions:**

The paper studies a distributed bandit setting where the goal is to maximize reward when 1) only a subset of the users participate in the learning process at every step 2) we also want to ensure some form of differential privacy (central, local, shuffle). The main contribution is differentially private distributed phased elimination (DP-DPE), which combines standard techniques for ensuring privacy + a variant of the usual phased elimination algorithm for linear bandits. The tradeoff that arises is between regret, privacy, and communication complexity.

**Broader Impact Concerns:**

I don’t see any broader impact concerns; this paper is on the theoretical end of the spectrum and in its current form does not suggest any potential broader impact concerns.

**Requested Changes:**

Important points:
- I think that the missing pieces discussed in Remark 1 and Remark 2 should be incorporated into the analysis; these are standard ideas and for completeness I think they should be reflected in the final statements.
- The experiments should be rerun with more realistic regimes of epsilon and delta parameters.

Minor points:
- It would be nice to strengthen the regret guarantees to hold with high probability and not just in expectation. The usual LinUCB analysis allows this, and indeed this seems implicit in your results.
- Instead of assuming that the rewards are bounded by B, it might be nicer to explicitly truncate them in the algorithm.
- I think the English should be revised in a few places just for the sake of correctness (the writing is already very clear; the instances I am referring to did not affect understanding).

**Strengths And Weaknesses:**

Strengths:
1. The exact setup of this paper is novel to the best of my knowledge, and it is reasonably well-motivated.
2. The writing is clear and flows nicely.
3. The analysis of the privacy-preserving portion of the algorithm and the bandit subroutine are largely independent, which facilitates exposition and understanding.

Weaknesses:
1. Although the exact setup seems novel, Algorithm 1 and its analysis seem to be a black-box combination of standard tools in bandits and privacy. Indeed, I don’t see any difficulty arising in putting the pieces together, as the confidence intervals simply get blown up due to privacy noise. On a related note, the noise arising from client-related noise at a technical level seems to be the same as having higher noise in the reward distribution, so this extra source of noise doesn’t seem to add much conceptual or technical novelty. Am I missing something here?
2. Section 6 was a bit odd. First, I think it’s misleading to say that privacy comes “for free”, as the cost incurred due to privacy can clearly dominate the other, privacy-independent cost. Second, the connection to DP-SCO seems too vague, high-level, and not particularly surprising. I didn’t feel like I got much insight from reading that section.
3. The experiments are not very thorough, and the choice of privacy parameters in some of them is very unusual (eps = 10, delta = 1/4 in Fig 2b, which is huge).  Is delta = 1/4 in Fig 2a as well? It would be good to show Fig 2b for different values of eps/delta.

---

### Review · Reviewer_AJwg · 2022-05-03

**Summary Of Contributions:**

The authors study a linear bandit problem with a distributed observation structure and privacy constraints. Namely, the learner selects action $x$ and a subset of users from a set of users, each user $u$ collects a noisy reward of mean $x^\top \theta_u$. It is assumed that for each $u$, $\theta_u$ is drawn from some distribution with mean $\theta^*$. The goal of the learner is to figure out the optimal action maximizing $x^\top \theta^*$, and users can choose to send information to help the learner find the optimal action. The authors study the trade-off between regret and communication cost, i.e. how much information must be sent to the learner by the users in order to find the optimal action. Also, the authors consider an additional privacy constraint, where the information sent from the users to the learner must be sufficiently anonymous (in the sense of differential privacy, with several possible definitions).

The authors then propose a family of algorithms for this problem, under various assumptions as far as differential privacy is concerned. The provide regret bounds for both regret and communication cost. They validate their theoretical findings with numerical simulations.



**Broader Impact Concerns:**

Not applicable.

**Requested Changes:**

* Clarify whether or not the set of users is finite, and the impact this has on the results
* Clarify how computationally difficult is the computation of the exploration strategy
* Quantify the communication cost in bits, as opposed to real numbers
* Comment on the information theoretic limits, if applicable

**Strengths And Weaknesses:**

Strengths:
* The problem at hand seems novel, and the authors provide an almost complete contribution with algorithms, regret bounds and numerical experiments
* The paper is very well written and easy to follow

Weaknesses

Size of the set of users: Unless I missed something, I believe that there is a problem with the model, specifically concerning the assumptions made on the set of users. The authors do not specify whether or not the set of users ${\cal U}$ is finite. Now if ${\cal U\}$ is finite there are some problems, for instance:
* the best action is not identifiable: since we assume that $\theta_u$ for $u \in {\cal U}$ are i.i.d samples from $\theta^*$, unless there are infinitely many users, one cannot estimate $\theta^*$ with arbitrary accuracy from a finite number of $\theta_u$ $u \in {\cal U}$. So it is not possible to identify the optimal action in general, and the regret of any algorithm should be linear $O(T)$.
* if the size of ${\cal U}$ is finite the algorithm provided by the authors is not applicable, in the sense that one must sample subsets of users with size that grows in $T$ in an unbounded fashion.

On the other hand, if ${\cal U}$ is infinite:

* Unless I have missed something, it is not obvious why privacy should be a concern, indeed, since if we have access to infinitely many users we can just sample a new user at every time step (the authors could comment on this).
* One can also wonder how applicable this is to systems like wireless networks which are used to model the problem, since the number of active users will at times be quite small in such systems
* If ${\cal U}$ is infinite, can one simply reduce the model to a problem where one can observe i.i.d. rewards with mean $x^\top \theta^\star$, by treating $x^\top (\theta^* - \theta_u)$ as noise ?

Exploration Strategy: When selecting the exploration strategy in the algorithm the authors specify that one can find a distribution $\pi_l(.)$ over ${\cal D}_t$ such that $g(\pi_l) \le 2d$. While I understand that it can be done, is it easy to do computationally ? For instance can this be done in time polynomial in $d$ ? Can this be done when the set of possible actions $x$ has a complicated structure ? For completeness, providing a subroutine for doing this could be good.

Communication Model: I have some doubts about two aspects of the communication model and how communication is performed by the algorithm:
-  one can send real numbers to the learner, which is clearly unrealistic and allows for an infinite amount of information to be transmitted (the authors even comment on this in the paper). Now I might be wrong but I believe that this can be fixed, in the sense that if a user wants to send an estimate $\hat \theta$ to the learner, it is enough to send him an approximate version $\tilde \theta$ instead, where $||\hat \theta - \tilde \theta || \le O(1/T^2)$, and one can do this by quantizing $\hat \theta$, so that only $O(d \log T)$  bits have to be sent. Still this should be investigated.
- the algorithm does not really attempt to minimize communication, and simply sends empirical average rewards for each action. Is this really necessary ? For instance it seems obvious that quantizing the empirical average rewards with a number of bits that depend on the phase would be better (since the accuracy we seek increases at each phase).

Information Theoretic Limits: The authors do not provide information theoretic lower bounds in order to quantify how far from optimal the proposed algorithms stand. For instance, it is understood that for sublinear regret the learner requires at least $O(d \log(T))$ bits of information, since he needs to be able to estimate $\theta^\star$ up to an error scaling as $o(T^{\alpha})$, where $\alpha > 0$. Now the communication cost is much larger than that for the proposed algorithm. So it would be essential to understand how efficient the proposed algorithm can be with respect to lower bounds.

---

### Review · Reviewer_HyQ1 · 2022-05-04

**Summary Of Contributions:**

In short, this paper proposes a novel distributed linear bandits problem and gives a differentially private algorithmic framework which achieves sub-linear regret. Specifically, the learner wants to learn the best vector $x \in D$ that maximizes reward $\langle  x, \theta^* \rangle$ for an unknown parameter $\theta^*$. In the distributed setting this paper concerns, after the learner chooses an action $x$ feedbacks the learner can get access to are so-called local rewards: the learner can samples some clients $u$ and the clients can locally learn $\langle x, \theta_u \rangle + \epsilon$, where $\theta_u$ is a sub-gaussian random vector with mean $\theta^*$ and $\epsilon$ is the noise term. Since feedbacks are distributed, any information transmitted from clients to the learner will incur communication cost. The objective is twofold: the learner wants to minimize the regret with a small communication cost.
The main contribution of this paper is a unified algorithmic framework DP-DPE that integrates three common differential privacy models (DP, local DP, and shuffle DP). DP-DPE is an extension of phased elimination algorithm for traditional stochastic linear bandits by Lattimore et al.
Authors also provide numerical simulation results.

**Broader Impact Concerns:**

I don't see potential ethical concerns of this work.

**Requested Changes:**

1. Adding lower bounds is important for acceptance as far as I am concerned. Considering that this paper does not provide much new insight regarding the upper bound part, I'm afraid that this submission does not have enough technical contribution without a lower bound discussion.

2. It would be great to discuss the importance of the assumption that $\theta_u$ is a sub-gaussian random vector with mean $\theta^*$. This requirement is unnecessarily strong in my eye. It's worth considering the case that drops the sub-gaussian condition.

**Strengths And Weaknesses:**

Strengths:
1. This work proposes a novel online learning setting that might be useful in varies applications.
2. It is very natural to consider differential privacy in the distributed setting. Authors give a unified framework that integrates common tools in different DP models. It is also nice to see this framework can protect privacy with minor impact on performance.
3. The paper is well-written, I appreciate intuitions and motivations provided by authors.

Weaknesses:
1. This work does not provide any discussion and/or proofs of lower bounds. We do not know whether DP-DPE (nearly) achieves the best tradeoffs between the regret and the communication cost.
2. The technical contribution is weak, I feel DP-DPE is a simple and natural extension of the traditional phased elimination algorithm. Compared with the traditional linear bandits setting, in my opinion, this new model only add one more layer of noise (distributed local reward). It is not surprising that one can easily adapt linear bandits algorithms to this distributed setting by adding one more step of sampling.

---

### Note · Authors · 2022-05-09

I have read and agree with the venue's withdrawal policy on behalf of myself and my co-authors.